# INFORMATIVE DATA SELECTION FOR THORAX DISEASE CLASSIFICATION

## ABSTRACT

Although Deep Neural Networks (DNNs) such as Vision Transformers (ViTs) have demonstrated superior performance in medical imaging tasks, the training of DNNs usually requires large amounts of high-quality labeled training data, which is usually difficult or even impractical to collect in the medical domain. To address this issue, Generative Data Augmentation (GDA) has been employed to improve the performance of DNNs trained on augmented training data comprising both original training data in the standard benchmark datasets and synthetic training data generated by generative models such as Diffusion Models (DMs). However, the synthetic data generated by GDA universally suffer from noise, and such synthetic data can severely hurt the performance of classifiers trained on the augmented training data. Existing works, such as data selection and data re-weighting methods aiming to mitigate this issue, usually depend on a given clean metadata or external classifier. In this work, we propose a principled sample re-weighting method, Informative Data Selection (IDS), based on an established information theoretic measure, the Information Bottleneck (IB), to improve the performance of DNNs trained for thorax disease classification with GDA. Extensive experiments demonstrate that IDS successfully assigns higher weights to more informative synthetic images and significantly outperforms existing data selection and data re-weighting methods in GDA for thorax disease classification. The code of IDS is available at https://anonymous.4open.science/r/IDS-20D1.

## 1 INTRODUCTION

Recent studies have pushed forward the development of deep neural networks (DNNs) for applications in medical imaging, such as disease classification for chest X-rays (Guendel et al., 2018; Xiao et al., 2023). Pioneering efforts utilized convolutional neural networks (CNNs), like U-Net (Ronneberger et al., 2015), to foster representation learning from radiography images. Lately, Vision Transformers (ViTs) (Dosovitskiy et al., 2020) have also been employed to harvest informative medical representations from these images (Xiao et al., 2023), leveraging their proficiency in handling long-range dependencies among features. While CNNs and ViTs have achieved promising results, their effectiveness largely depends on the quality and volume of the available data and annotations (Feng et al., 2020). However, collecting a large dataset of high-quality annotations in medical domains is notably challenging (El Jiani et al., 2022; Xiao et al., 2023) or even impractical (Esteva et al., 2021; Price & Cohen, 2019; Ali et al., 2023; Ramudu et al., 2023) due to resource limitations or privacy issues. To overcome this issue, self-supervised learning (SSL), such restorative learning (Xiao et al., 2023), has been utilized to procure representations from unlabeled data. More recently, following the success of generative models (Rombach et al., 2022; Akrout et al., 2023), generative data augmentation (GDA) (Sarıyıldız et al., 2023; Lei et al., 2023; Azizi et al., 2023b; Trabucco et al., 2024a), aiming to synthesize labeled training data using deep generative models, has also emerged as a potent strategy to enrich training datasets.

**Generative Data Augmentation (GDA) for Disease Classification.** Data scarcity and the lack of high-quality labeled training data is a long-standing challenge in medical imaging and also general computer vision. To address this issue, the literature has conducted extensive studies in GDA (Sarıyıldız et al., 2023; Lei et al., 2023; Azizi et al., 2023b; Trabucco et al., 2024a), such as that based on Generative Adversarial Networks (GANs) (Zhang et al., 2021; Li et al., 2022) and Diffusion Models (DMs) (He et al., 2023b; Tian et al., 2023; Yuan et al., 2022; Bansal & Grover, 2023; Vendrow et al., 2023), which have demonstrated promising results in applications in both

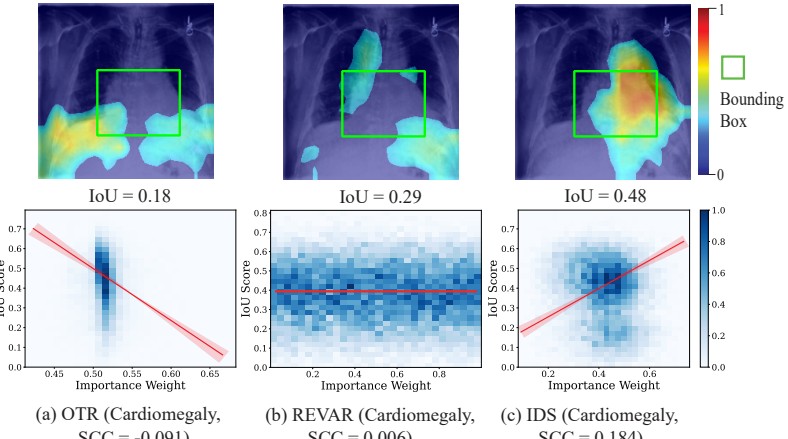

Figure 1: **Figures in the first row** illustrate examples of thresholded Grad-CAM visualization for OTR, REVAR and IDS. For each of the examples, we also present the ground-truth bounding box for the disease. The thresholded heatmap areas are considered as the disease localization areas. IoU score between the disease localization area and the ground-truth bounding box is shown below each example. A synthetic image with a higher IoU score is considered a more informative sample for this disease as a larger portion of the predicted disease localization area overlaps with the ground-truth bounding box of the disease. **Figures in the second row** illustrate the correlation between IoU scores for disease localization and importance weights for OTR (Guo et al., 2022), REVAR (Jain et al., 2024), and IDS in the CheXpert dataset. The disease name and Spearman Correlation Coefficients (SCC) (Spearman, 1961) are attached in the parenthesis. A larger absolute value of a positive SCC between two variables indicates a stronger positive correlation, which refers to a correlation between two variables where as one variable increases, the other variable tends to increase as well. The range of IoU and the range of the importance weight, which is $[0, 1] \times [0, 1]$, is divided into $30 \times 30$ cells evenly, and the color of each cell is proportional to the number of synthetic images whose IoU sores and importance weights fall in that cell. As a result, a cell with more blue indicates more synthetic images falling in that cell. The red lines in the figures are the linear regression results between the IoU scores and the importance weights, which visualizes the correlation. It can be observed that the linear regressors in red suggest a stronger positive correlation between the IoU scores and the importance weights by our IDS than that for competing baselines, which is further quantitatively evidenced by the higher SCC for IDS than the competing baselines. The correlation analysis on NIH ChestX-ray14 is illustrated in Figure 4 in Section D.2 of the appendix.

general computer vision (Sarıyıldız et al., 2023; Azizi et al., 2023b; Trabucco et al., 2024a) and medical imaging, such as medical image classification (Akrout et al., 2023) and medical anomaly detection (Wolleb et al., 2022). Motivated by this observation, this paper aims to improve the performance of DNNs trained for thorax disease classification with the augmented training data comprising both original training images in the benchmark datasets and synthetic images generated by DMs.

**Challenges in GDA for Disease Classification.** Albeit the potential of GDA, the synthetic data generated by GDA universally suffer from noise (He et al., 2023a; Azizi et al., 2023a), and such synthetic data can severely hurt the performance of classifiers trained on the augmented training data. To address this issue, the literature widely adopts data selection (Chhabra et al., 2024) or sample re-weighting methods (He et al., 2023a), which use re-weighted or selected synthetic data when training the classifier. Existing sample re-weighting methods (Shu et al., 2019; Guo et al., 2022; Jain et al., 2024) typically depend on training a meta-network using certain clean metadata, hoping that such a network can assign higher importance weights to more informative training samples. However, all these methods assume the existence of such clean metadata, and it is not clear how such metadata can be obtained for the medical task considered in this paper without efforts from medical experts. The existing work that is the closest to our setup is CBF (He et al., 2023a), which introduces a CLIP Filter strategy to rule out noisy synthetic data. The CLIP Fiter employs CLIP zero-shot classification confidence to assess the quality of the synthesized data, and the synthetic data with low-confidence are filtered out. The performance of CBF highly depends on the zero-shot image classification capability of a vision-language model, CLIP (Radford et al., 2021), which is pre-trained on a huge dataset of image and text pairs. However, as a method highly depending a vision-language model

pre-trained on generic data, CLIP may not be able to render reliable predictions on the specialized data, such as the X-rays for thorax disease classification considered in this paper.

In summary, the current medical imaging and general machine learning literature lack a principled sample re-weighting method for GDA which does not depend on a given clean metadata or external classifiers. The major contribution of this paper is a principled sample re-weighting method based on an established information theoretic measure, the Information Bottleneck (IB), which does not require either clean metadata or external classifiers and exhibits superior performance over the competing methods under the GDA setup for thorax disease classification.

**Our Contributions.** We propose a principled sample re-weighting method, Informative Data Selection (IDS), for training DNNs with GDA, which assigns higher importance weights to more informative samples based on an IB theoretic framework. The detailed contributions of this paper are presented as follows.

First, to the best of our knowledge, IDS is among the first to perform sample re-weighting in GDA by employing a principled IB framework in the sample re-weighting process where each synthetic image receives an importance weight, aiming to improve the accuracy of the classifier trained on the augmented data comprising the original training data and the re-weighted synthetic data. In contrast to existing works in sample re-weighting (Shu et al., 2019; Guo et al., 2022; Jain et al., 2024) and data selection (Chhabra et al., 2024; He et al., 2023a), IDS does not require either clean metadata or external classifiers.

Second, the sample re-weighting network is optimized for reducing the IB loss on the synthetic data, such that the IB principle, learning features more strongly correlated with class labels while decreasing their correlation with the inputs, is better adhered. To achieve this goal, the importance weights generated by the sample re-weighting network are applied to the input features and representations of the synthetic data to compute weighted class centroids in both the input feature space and representation space, which are then used to compute the IB loss on the synthetic data. To minimize the IB loss with minibatch-based SGD algorithms, we further derive a separable variational upper bound of the IB loss, termed the VIB. In the training process, the cross-entropy loss re-weighted by the importance weights and the VIB are iteratively optimized to update the weights in the classification network and the sample re-weighting network. As evidenced by the results in Section 4.2, IDS significantly outperforms state-of-the-art data re-weighting (Shu et al., 2019; Guo et al., 2022; Jain et al., 2024) and data selection methods (He et al., 2023a; Chhabra et al., 2024) for thorax disease classification on three thorax disease classification benchmarks, CheXpert (Irvin et al., 2019), COVIDx (Pavlova et al., 2022), and NIH ChestX-ray14 (Wang et al., 2017), demonstrating the superiority of IDS in selecting informative synthetic data for GDA.

To demonstrate the superiority of IDS in selecting informative samples, we study the correlation between the Intersection over Union (IoU) score for disease localization and importance weights learned by the baseline sample re-weighting methods (Guo et al., 2022; Jain et al., 2024) and IDS. The IoU score for disease localization is computed between the disease localization area predicted by our IDS and the competing baselines and the ground-truth bounding box of the disease in the X-ray images. Examples of disease localization areas are illustrated in the first row of Figure 1. A synthetic image with a higher IoU score between the disease localization area and the ground-truth bounding box is considered a more informative sample for this disease because a higher IoU means a larger portion of the predicted disease localization area overlaps with the ground-truth bounding box of the disease. More details on the ablation study can be found in Section 4.3 of the paper. The Spearman Correlation Coefficient (SCC) (Spearman, 1961) is used to quantitatively measure the correlation, and a larger absolute value of a positive SCC indicates a stronger positive correlation. Both quantitative and visualization results in Figure 1 illustrate a stronger positive correlation between the IoU scores and the importance weights learned by our IDS than that for competing baselines, demonstrating the superiority of IDS for assigning higher importance weights to more informative synthetic images to improve the accuracy of the classifier trained on the augmented data.

## 2 RELATED WORKS

### 2.1 MEDICAL IMAGE ANALYSIS WITH DEEP LEARNING

Deep learning has made remarkable progress in photographic image analysis (Lin et al., 2017b;a), sparking interest in applying it to medical imaging. Convolutional neural networks (CNNs) like U-Net (Falk et al., 2018; Zhou et al., 2018) pioneered this field, achieving state-of-the-art performance

across various tasks such as image classification (Wang et al., 2019; Ma et al., 2020), object detection (Falk et al., 2019; Zhou et al., 2018; Yang & Yu, 2021), and semantic segmentation (Yang & Yu, 2021; Yao et al., 2021). More recently, vision transformers, inspired by the success of transformers in natural language processing (Vaswani et al., 2017), have outperformed state-of-the-art CNNs on various computer vision benchmarks (Zhu et al., 2021; Cai et al., 2023). Their self-attention mechanism can better model long-range dependencies compared to CNNs' local convolutions (Li et al., 2023b). Given the scarcity of high-quality annotations, self-supervised contrastive learning techniques (Chen et al., 2020a; Caron et al., 2020; Xiao et al., 2023) have gained traction for pre-training networks in this domain (Zhou, 2021; Xiao et al., 2023; Chen et al., 2021). However, the high similarity between radiographic images due to standardized protocols (Xiang et al., 2021; Haghighi et al., 2022) poses challenges compared to photographic images (He et al., 2020; Chen et al., 2020b). To address this, recent works utilize restorative strategies like masked autoencoders (MAE) (He et al., 2022) for pre-training (Xiao et al., 2023).

## 2.2 Existing Works about Information Bottleneck

The Information Bottleneck (IB) principle (Tishby et al., 2000) posits that an optimal DNN compresses its input data, preserving only the information that is essential for predicting the target outputs, thereby maximizing the mutual information between the representations and the target outputs while minimizing the mutual information between the input and the representations. Deep VIB (Alemi et al., 2017) first integrates the IB principle as a training objective for deep neural networks. Both empirical (Lai et al., 2021; Zhou et al., 2022) and theoretical (Amjad & Geiger, 2020; Kawaguchi et al., 2023) works prove that DNNs better adhering to the IB principle show stronger performance. In the medical imaging domain, the IB principle is also widely adopted to learn discriminative task-oriented image representations (Demir et al., 2021; Wang et al., 2023; Schott et al., 2024; Li et al., 2023a). MIB-Net (Wang et al., 2023) multiplies a contribution score map with the input image to force the network to learn representations that are more correlated with the target task. IB-TransUNet (Li et al., 2023a) introduces an information bottleneck block in to compress redundant features and reduce the risk of overfitting in medical image segmentation tasks. In contrast with existing works that utilize the IB principle to enhance the image representation learning capabilities of DNNs, our method is the first that utilizes the IB principle for selecting high-quality synthetic data to augment the training of DNNs for the medical image classification task.

## 2.3 Existing Works about Generative Data Augmentation, Data Selection and Sample Re-weighting

Generating synthetic informative training data as data augmentation, or generative data augmentation (GDA), for improving the performance of DNNs remains a vital yet challenging research area. Existing works (Sarıyıldız et al., 2023; Lei et al., 2023; Azizi et al., 2023b; Trabucco et al., 2024a) predominantly focus on synthesizing training data through deep generative models, such as Generative Adversarial Networks (GANs) (Zhang et al., 2021; Li et al., 2022) and diffusion models (He et al., 2023b; Tian et al., 2023; Yuan et al., 2022; Bansal & Grover, 2023; Vendrow et al., 2023). In the medical domain, researchers have also explored employing generative models to synthesize training images for data augmentation, addressing the lack of high-quality labeled data (Shin et al., 2018; Zhu et al., 2017; Jiang et al., 2018; Sharma & Hamarneh, 2019; Cha et al., 2020; Akrout et al., 2023) in tasks such as medical image classification (Shin et al., 2018) and medical anomaly detection (Akrout et al., 2023). Despite works showing that synthetic images can improve the training of DNNs for medical tasks, they often overlook the fact that synthetic data produced by generative models can introduce noise (Azizi et al., 2023b; Trabucco et al., 2024b; Na et al., 2024), which underscores the critical need for careful quality control in using the generated synthetic data for data augmentation. To mitigate this issue, recent works focus on three directions: improving the quality of the synthetic data, data selection, and data re-weighting. The first category of methods seeks to directly improve the quality of the generated synthetic data by refining the generation process of diffusion models to (Sarıyıldız et al., 2023; Lei et al., 2023; Zhou et al., 2023). The second category of methods, data selection (Wu et al., 2021; Nguyen et al., 2020; Song et al., 2023; Lin et al., 2023; He et al., 2023a; Chhabra et al., 2024), which aims to select a high-quality subset from the noisy training data to improve the performance of deep learning models, can also be used to select high-quality synthetic data. For example, Classifier-based Filtering (CBF) (He et al., 2023a) proposes to select synthetic images with high CLIP zero-shot classification confidence. The third category, data re-weighting, uses soft data selection by assigning importance weights to training samples (Mo et al., 2019; Shu et al., 2019; Guo et al., 2022; Jain et al., 2024). Methods like Meta-Weight-Net

(Shu et al., 2019), OTR (Guo et al., 2022), and REVAR (Jain et al., 2024) employ meta-learning to adaptively learn sample weights from a clean meta dataset, enhancing robustness against noise or bias in the training data (Shu et al., 2019; Guo et al., 2022; Jain et al., 2024).

# 3 INFORMATIVE DATA SELECTION

Given the original training set $\mathcal{D}_{\text{real}} = \{x_i, y_i\}_{i=1}^N$ for Thorax disease classification, we aim to generate synthetic training set $\mathcal{D}_{\text{syn}} = \{\widehat{x}_j, \widehat{y}_j\}_{j=1}^M$ with diffusion models and train a classifier on the augmented training set $\mathcal{D}_{\text{aug}} = \mathcal{D}_{\text{real}} \cup \mathcal{D}_{\text{syn}}$. To mitigate the negative effects of potential abundant noise in the synthetic training samples, we propose Informative Data Selection (IDS) to re-weight the synthetic training samples with a sample re-weighting network. The sample re-weighting network is trained by minimizing the variational upper bound for the Informative Bottleneck (IB) loss on the synthetic training set in the hope that more informative synthetic training samples can have higher weights, thus improving the performance of the classifier trained on the augmented training set.

In Section 3.1, we first describe the details for generating the synthetic training samples with diffusion models. Next, we derive the variational upper bound for the IB loss in Section 3.2. In Section 3.3, we describe the training of the re-weighting network and the classifier network in IDS.

## 3.1 GENERATING SYNTHETIC TRAINING SAMPLES WITH DIFFUSION MODELS

To generate labeled synthetic training samples, we train a conditional Latent Diffusion Model (LDM) (Rombach et al., 2022) with Classifier-Free Guidance (CFG) (Ho & Salimans, 2022) on the latent features of the images in the training set, which are generated by an off-the-shelf pre-trained variational autoencoder (VAE) model from Stable Diffusion (Rombach et al., 2022). Detailed formulations of the training and inference of diffusion models, LDM, and CFG are deferred to Section B.1 of the appendix. We use Diffusion Transformers (DiTs) (Peebles & Xie, 2023) as the backbones of the LDMs in our works. Let $v_{\text{e}}$ and $v_{\text{d}}$ be the fixed pre-trained encoder and decoder. The encoder of the VAE is first applied to generate the latent features $\{h_i\}_{i=1}^N$ of $\mathcal{D}_{\text{real}}$, where $h_i = v_{\text{e}}(x_i)$ is the latent feature of the $i$-th image. The parameters of the LDM $\omega$ are trained on $\{h_i, y_i\}_{i=1}^N$ by minimizing the loss $\mathcal{L}_{\text{LDM}}$ in Equation (18) in Section B.1 of the appendix. Algorithm 1 in Section B.1 of the appendix describes the training algorithm of the LDM.

Once the training of the LDM is finished, the latent features $\left\{\widehat{h}_j\right\}_{j=1}^M$ are generated for a set of pre-defined synthetic labels $\{\widehat{y}_j\}_{j=1}^M$ using Equation (17) in Section B of the appendix. The synthetic training images $\{\widehat{x}_j\}_{j=1}^M$ are then generated by applying the pre-trained decoder on the latent features $\left\{\widehat{h}_j\right\}_{j=1}^M$, where $\widehat{x}_j = v_{\text{d}}\left(\widehat{h}_j\right)$. In our work, we set the synthetic labels $\{\widehat{y}_j\}_{j=1}^M$ to be the same as the original label set $\{\widehat{y}_j\}_{j=1}^M$. Algorithm 2 in Section B.1 of the appendix describes the generation process of the synthetic training set. After obtaining the synthetic training set $\mathcal{D}_{\text{syn}} = \{\widehat{x}_j, \widehat{y}_j\}_{j=1}^M$ with the LDM, we can combine it with the original training set $\mathcal{D}_{\text{real}}$ to obtain the augmented training set $\mathcal{D}_{\text{aug}} = \mathcal{D}_{\text{real}} \cup \mathcal{D}_{\text{syn}}$. Next, the classifier network in IDS can be trained together with the sample re-weighing network on $\mathcal{D}_{\text{aug}}$ as described in Section 3.3.

## 3.2 VARIATIONAL UPPER BOUND FOR THE IB LOSS

In order to assign higher importance weights to more informative synthetic training samples, we propose to train the re-weighting network by minimizing the IB loss on the synthetic training set. To achieve this goal, we first derive a variational upper bound for the IB loss, which can be optimized by standard SGD algorithms. Given the synthetic training set $\mathcal{D}_{\text{syn}} = \{\widehat{x}_j, \widehat{y}_j\}_{j=1}^M$, we first specify how to compute the IB loss, $\text{IB}(\Theta) = I(\widehat{Z}(\Theta), \widehat{X}) - I(\widehat{Z}(\Theta), \widehat{Y})$, where $\Theta$ is the weights of a neural network, $\widehat{X}$ is a random variable representing the input feature of the synthetic training sample, which takes values in $\{\widehat{x}_j\}_{j=1}^M$, $\widehat{Z}(\Theta)$ is a random variable representing the learned feature of the synthetic training sample, which takes values in $\{\widehat{z}_j(\Theta)\}_{j=1}^M$ with $\widehat{z}_j(\Theta)$ being the learned feature for the $j$-th synthetic training sample. $\widehat{Y}$ is a random variable representing the synthetic class label, which takes values in $\{y_j\}_{j=1}^n$. We define $\mathcal{C}(\theta, \Theta) = \left\{ \left\{ c_k^{(\text{input})}(\theta) \right\}_{k=1}^C, \left\{ c_k^{(\text{feat})}(\theta, \Theta) \right\}_{k=1}^C \right\}$ as the class cen-

troids of the input features and the learned features on the synthetic training set, where $\theta$ denotes the parameters of the sample re-weighting network. The formulas for the computation of $\mathcal{C}(\theta, \Theta)$ can be found in Equation (3). We abbreviate $\widehat{Z}(\Theta)$ as $\widehat{Z}$, $c_k^{(\text{input})}(\theta)$ as $c_k^{(\text{input})}$, and $c_k^{(\text{feat})}(\theta, \Theta)$ as $c_k^{(\text{feat})}$ for simplicity of the notations. Then we define the probability that $\widehat{z}_j$ belongs to class $a$ as $\Pr\left[\widehat{Z} \in a\right] = \frac{1}{M} \sum_{j=1}^{M} \phi(\widehat{z}_j, c_a^{(\text{feat})})$ with $\phi(\widehat{z}_j, c_a^{(\text{feat})}) = \frac{\exp\left(-\|\widehat{z}_j - c_a^{(\text{feat})}\|_2^2\right)}{\sum_{a=1}^{C} \exp\left(-\|\widehat{z}_j - c_a^{(\text{feat})}\|_2^2\right)}$. Similarly, we define the probability that $\widehat{x}_j$ belongs to class $b$ as $\Pr\left[\widehat{X} \in b\right] = \frac{1}{n} \sum_{j=1}^{M} \phi(x_j, c_b^{(\text{input})})$. Moreover, we have the joint probabilities $\Pr\left[\widehat{Z} \in a, \widehat{X} \in b\right] = \frac{1}{M} \sum_{j=1}^{M} \phi(\widehat{z}_j, c_a^{(\text{feat})}) \phi(\widehat{x}_j, c_b^{(\text{input})})$ and $\Pr\left[\widehat{Z} \in a, \widehat{Y} = y\right] = \frac{1}{M} \sum_{j=1}^{M} \phi(\widehat{z}_j, c_a^{(\text{feat})}) \mathbb{1}_{\{\widehat{y}_i = y\}}$ where $\mathbb{1}_{\{\}}$ is an indicator function. As a result, we can compute the mutual information $I(\widehat{Z}, \widehat{X}) = \sum_{a=1}^{C} \sum_{b=1}^{C} \Pr\left[\widehat{Z} \in a, \widehat{X} \in b\right] \log \frac{\Pr[\widehat{Z} \in a, X \in b]}{\Pr[\widehat{Z} \in a] \Pr[\widehat{X} \in b]}$, $I(\widehat{Z}, \widehat{Y}) = \sum_{a=1}^{C} \sum_{y=1}^{C} \Pr\left[\widehat{Z} \in a, \widehat{Y} = y\right] \log \frac{\Pr[\widehat{Z} \in a, \widehat{Y} = y]}{\Pr[\widehat{Z} \in a] \Pr[\widehat{Y} = y]}$, and then compute the IB loss $\text{IB}(\mathcal{C}(\theta, \Theta), \Theta, \mathcal{D}_{\text{syn}})$. Given a variational distribution $Q(\widehat{Z} \in a | Y = y)$ for $y \in \{1, \ldots, C\}$ and $a \in \{1, \ldots, C\}$, the following theorem gives a variational upper bound, $\text{VIB}(\mathcal{C}(\theta, \Theta), \Theta, \mathcal{D}_{\text{syn}})$, for the IB loss $\text{IB}(\mathcal{C}(\theta, \Theta), \Theta, \mathcal{D}_{\text{syn}})$.

**Theorem 3.1.**

$$\text{IB}(\mathcal{C}(\theta, \Theta), \Theta, \mathcal{D}_{\text{syn}}) \leq \text{VIB}(\mathcal{C}(\theta, \Theta), \Theta, \mathcal{D}_{\text{syn}}), \tag{1}$$

where

$$\text{VIB}(\mathcal{C}(\theta, \Theta), \Theta, \mathcal{D}_{\text{syn}}) := \frac{1}{M} \sum_{j=1}^{M} \text{VIB}(\mathcal{C}(\theta, \Theta), \Theta, \widehat{x}_j), \tag{2}$$

$$\text{VIB}(\mathcal{C}(\theta, \Theta), \Theta, \widehat{x}_j) := \sum_{a=1}^{C} \sum_{b=1}^{C} \phi(\widehat{z}_j, c_a^{(\text{feat})}) \phi(\widehat{x}_j, c_b^{(\text{input})}) \log \phi(\widehat{x}_j, c_b^{(\text{input})})$$

$$- \sum_{a=1}^{C} \sum_{y=1}^{C} \phi(\widehat{z}_j, c_a^{(\text{feat})}) \mathbb{1}_{\{\widehat{y}_j = y\}} \log Q(\widehat{Z} \in a | Y = y).$$

$\text{VIB}(\mathcal{C}(\theta, \Theta), \Theta, \widehat{x}_j)$ can be interpreted as the information bottleneck upper bound for the $j$-th synthetic image. The proof of this theorem follows by applying Lemma A.1 and Lemma A.2 in Section A of the supplementary. We remark that $\text{VIB}(\Theta)$ is ready to be optimized by standard SGD algorithms because it is separable and expressed as the summation of losses on individual training points. In order to compute $\text{VIB}(\Theta)$ before a new epoch starts, we need to update the variational distribution $Q^{(t)}$ at the end of the previous epoch.

### 3.3 Formulation of Informative Data Selection (IDS)

Given the original training set $\mathcal{D}_{\text{real}} = \{x_i, y_i\}_{i=1}^{N}$ and the synthetic training set $\mathcal{D}_{\text{syn}} = \{\widehat{x}_j, \widehat{y}_j\}_{j=1}^{M}$ generated by the diffusion model, we aim to train an image classifier $f_{\Theta}(\cdot)$ on the augmented training set $\mathcal{D}_{\text{aug}} = \mathcal{D}_{\text{real}} \cup \mathcal{D}_{\text{syn}}$, where $f_{\Theta}(\cdot)$ is a DNN and $\Theta$ denotes its network parameters. However, training the classifier directly on the augmented training set can hurt the performance of the classifier due to the potential abundant noise in the synthetic images in $\mathcal{D}_{\text{syn}}$. To address this issue, we train a sample re-weighting network $g_{\theta}(\cdot)$ to learn importance weights $\{g_{\theta}(\widehat{x}_j) \in [0, 1]\}_{j=1}^{M}$ for training samples in the synthetic training set $\mathcal{D}_{\text{syn}}$, where $g_{\theta}(\cdot)$ is a DNN and $\theta$ denotes its parameters. We remark that the re-weighting network plays a role similar to that of the meta networks in (Shu et al., 2019; Jain et al., 2024), which generate the importance weights for training samples.

To train the sample re-weighting network $g_{\theta}(\cdot)$, such that more informative samples in $\mathcal{D}_{\text{syn}}$ can have higher weights, we train $g_{\theta}(\cdot)$ by optimizing the variational upper bound of the IB loss, VIB, on the synthetic training set $\mathcal{D}_{\text{syn}}$. To compute the VIB on the synthetic training set $\mathcal{D}_{\text{syn}}$, we first

compute the class centroids for the input features and the image representations using all the images in the augmented training set $\mathcal{D}_{\text{aug}}$. Let $f'_\Theta(\cdot)$ denote the representation learning backbone of the image classifier $f_\Theta(\cdot)$ excluding the last linear layer. The class centroids for the input features and the image representations can be computed by

$$c_k^{(\text{input})}(\theta) = \frac{\sum_{i=1}^N x_i \mathbb{I}_{\{y_i=k\}} + \sum_{j=1}^M g_\theta(\widehat{x}_j)\widehat{x}_j \mathbb{I}_{\{\widehat{y}_j=k\}}}{\sum_{i=1}^N \mathbb{I}_{\{y_i=k\}} + \sum_{j=1}^M g_\theta(\widehat{x}_j)\mathbb{I}_{\{\widehat{y}_j=k\}}},$$

$$c_k^{(\text{feat})}(\theta, \Theta) = \frac{\sum_{i=1}^N x_i \mathbb{I}_{\{y_i=k\}} + \sum_{j=1}^M g_\theta(\widehat{x}_j)f'_\Theta(\widehat{x}_j)\mathbb{I}_{\{\widehat{y}_j=k\}}}{\sum_{i=1}^N \mathbb{I}_{\{y_i=k\}} + \sum_{j=1}^M g_\theta(\widehat{x}_j)\mathbb{I}_{\{\widehat{y}_j=k\}}}, \tag{3}$$

where $k \in [C]$ is the class index and $C$ is the number of classes. $\mathbb{I}_{\{\}}$ is an indicator function. Next, the VIB on the synthetic training set $\mathcal{D}_{\text{syn}}$ can be computed using Equation (3). With the sample re-weighting network $g_\theta(\cdot)$, the overall training loss for the classifier $f_\Theta(\cdot)$ on the augmented training set $\mathcal{D}_{\text{aug}}$ is

$$\mathcal{L}_{\text{train}}(\theta, \Theta, \mathcal{D}_{\text{aug}}) = \frac{1}{N}\sum_{i=1}^N \text{CE}\left(f_\Theta(x_i), y_i\right) + \frac{1}{M}\sum_{j=1}^M g_\theta(\widehat{x}_j)\text{CE}\left(f_\Theta(\widehat{x}_j), \widehat{y}_j\right), \tag{4}$$

where $\text{CE}(,)$ is the cross-entropy function. To train the classifier $f_\Theta(\cdot)$ by minimizing $\mathcal{L}_{\text{train}}(\theta, \Theta, \mathcal{D}_{\text{aug}})$ while training the sample re-weighting network $g_\theta$ by minimizing $\text{VIB}(\theta, \Theta, \mathcal{D}_{\text{syn}})$, we formulate a bi-level optimization objective for IDS as

$$\Theta^* = \arg\min_\Theta \mathcal{L}_{\text{train}}(\theta^*, \Theta, \mathcal{D}_{\text{aug}}), \text{ s.t. } \theta^* = \arg\min_\theta \text{VIB}(\mathcal{C}(\theta, \Theta^*), \Theta^*, \mathcal{D}_{\text{syn}}), \tag{5}$$

where $\Theta^*$ and $\theta^*$ are the optimal parameters for the classifier $f_\Theta(\cdot)$ and the sample re-weighting network $g_\theta(\cdot)$.

**Optimization of IDS.** To train the classifier $f_\Theta(\cdot)$ and the sample re-weighting network $g_\theta(\cdot)$ with the optimization objective in Equation (5), we adopt an alternating stochastic gradient descent update strategy commonly used for solving bi-level optimization problems (Shu et al., 2019; Algan & Ulusoy, 2021; Jain et al., 2024). This process alternates between updating weights and classifier parameters, leveraging gradient-based methods to efficiently manage the interdependencies between the two tasks. In the bi-level optimization framework used here, the lower level optimizes a sample re-weighting network to assign importance weights to training samples, enhancing classifier training. The upper level then trains the classifier with these weighted samples for improved generalization. At the $t$-th epoch, the parameters of the sample re-weighting network are first updated by

$$\theta^{(t)} = \theta^{(t-1)} - \eta_\theta \nabla_\theta \text{VIB}(\mathcal{C}(\theta, \Theta^{(t-1)}), \Theta^{(t-1)}, \mathcal{D}_{\text{syn}}), \tag{6}$$

where $\eta_\theta$ is the learning rate of $\theta$. $\theta^{(t)}$ and $\Theta^{(t)}$ are the parameters of the sample re-weighting network and the classifier network at the $t$-th epoch. Next, the parameters of the classifier are updated by

$$\Theta^{(t)} = \Theta^{(t-1)} - \eta_\Theta \nabla_\Theta \mathcal{L}_{\text{train}}(\theta^{(t-1)}, \Theta, \mathcal{D}_{\text{aug}}), \tag{7}$$

where $\eta_\Theta$ is the learning rate of $\Theta$. Since both VIB and $\mathcal{L}_{\text{train}}$ are separable and amenable to mini-batch stochastic gradient descent (SGD), the entire optimization process of IDS can be efficiently conducted using mini-batch SGD. Algorithm 3 in Section C of the appendix describes the training process of IDS.

We remark that IDS can be easily extended to multi-label classification tasks. Let $L$ be the number of labels. The sample re-weighting network $g_\theta(\cdot)$ learns importance weight vectors $\left\{g_\theta(\widehat{x}_j) \in [0,1]^L\right\}_{j=1}^M$ for training samples in the synthetic training set $\mathcal{D}_{\text{syn}}$, where the $l$-th element of $g_\theta(\widehat{x}_j)$ corresponds to the importance of the $j$-th synthetic training sample for the $l$-th label. The training loss in Equation (4) and the VIB in Equation (2) can be separately computed for each of the $L$ labels. Let $\mathcal{L}_{\text{train}}(\theta, \Theta, \mathcal{D}_{\text{aug}}, l)$ and $\text{VIB}(\mathcal{C}(\theta, \Theta), \Theta, \mathcal{D}_{\text{syn}}, l)$ be the training loss and VIB corresponds to the $l$-th label. The parameters of the classification network and the sample re-weighting network can be optimized by replacing the training loss and VIB in the bi-level optimization objective in Equation (5) with $\frac{1}{L}\sum_{l=1}^L \mathcal{L}_{\text{train}}(\theta, \Theta, \mathcal{D}_{\text{aug}}, l)$ and $\frac{1}{L}\sum_{l=1}^L \text{VIB}(\mathcal{C}(\theta, \Theta), \Theta, \mathcal{D}_{\text{syn}}, l)$, respectively.

## 4 EXPERIMENTS

In this section, we present a comprehensive evaluation of our proposed Informative Data Selection (IDS) method across several medical imaging datasets. First, in Section 4.1, the implementation details of our experiments are presented. We perform the comparison between IDS and other data selection and sample re-weighting techniques on CheXpert, COVIDx, and NIH-ChestXray-14 in Section 4.2. In Section 4.3, we perform an ablation study to analyze the correlation between disease localization performance and importance weights for IDS and the baseline methods. In addition, the details for generating synthetic images with diffusion models are deferred to Section B.2 of the appendix. Additional experiment results are deferred to Section D of the appendix. Additional implementation details and experimental setups are presented in Section D.1 of the appendix. Additional ablation study results are presented in Section D.2. Finally, comparisons with more baseline methods for thorax disease classification across the three benchmarks are presented in Section D.7 of the appendix, and Grad-CAM visualization results on the NIH ChestXray-14 dataset are shown in Section D.8.

### 4.1 IMPLEMENTATION DETAILS

We evaluate the effectiveness of the proposed IDS method for thorax disease classification with base classification networks, ViT-S and ViT-B (Dosovitskiy et al., 2020), pre-trained on $266,340$ and $489,090$ chest X-rays with Masked Autoencoders (MAE) respectively, following the settings in (Xiao et al., 2023). After pre-training, the networks using the IDS are fine-tuned for three thorax disease classification datasets, CheXpert (Irvin et al., 2019), COVIDx (Pavlova et al., 2022), and NIH ChestX-ray14 (Wang et al., 2017). In addition to applying IDS for data re-weighting on the synthetic data, we also assess the performance of IDS for re-weighting both the real data and the synthetic data. More implementation details and experimental setups are deferred to Section D.1 of the appendix. The mean Area Under the Curve (mAUC) is adopted as the evaluation metric for the multi-label disease classification datasets CheXpert and NIH ChestX-ray14. The mAUC is computed by averaging the individual Area Under the Curve (AUC) values calculated for each disease label. Classification accuracy is used as the metric for the single-label dataset COVIDx.

### 4.2 EXPERIMENTAL RESULTS

**CheXpert.** Table 1 compares the performance of competing data selection and data re-weighting methods with our IDS for GDA on CheXpert. The base model ViT-B achieves a mAUC of $89.3\%$ when fine-tuned on the CheXpert dataset. By incorporating IDS for GDA, the IDS-ViT-B model attains a state-of-the-art mAUC of $90.1\%$, reflecting a $0.8\%$ improvement over the ViT-B and a $1.1\%$ improvement over the ViT-B trained with synthetic data. Notably, IDS models significantly outperform other data selection and data re-weighting methods for GDA. For instance, IDS-ViT-B outperforms REVAR by $0.8\%$ in mAUC. Moreover, applying IDS to re-weight both the real data and the synthetic data further boosts the performance of IDS. For example, IDS-ViT-B re-weighting both the synthetic data and the real data outperforms IDS-ViT-B re-weighting only the synthetic data by $0.6\%$ in mAUC, demonstrating the merits of IDS in selecting informative samples in both real data and synthetic data. Comparisons with additional baseline methods are provided in Table 8 in Section D.7 of the appendix.

Table 1: The performance of various state-of-the-art (SOTA) baseline methods on CheXpert. The best results are in bold, and the second-best results are underlined, for each architecture. Comparisons with more baselines are deferred to Table 8 in Section D.7 of the appendix.

| Method | Architecture | Atelectasis | Cardiomegaly | Edema | mAUC (%) |
|---|---|---|---|---|---|
| MAE (Xiao et al., 2023) | ViT-S/16 | 83.5 | 81.8 | 94.0 | 89.2 |
| MAE with Synthetic Data | | 83.0 | 81.5 | 94.0 | 88.6 |
| MW-Net (Shu et al., 2019) | | 81.7 | 82.7 | 94.1 | 88.9 |
| OTR (Guo et al., 2022) | | 84.6 | 81.2 | 94.2 | 89.0 |
| IE (Chhabra et al., 2024) | | 81.7 | 82.0 | 94.2 | 88.9 |
| CBF (He et al., 2023a) | | 81.4 | 82.7 | 94.2 | 88.8 |
| REVAR (Jain et al., 2024) | | 83.0 | 82.7 | 94.0 | 89.0 |
| IDS (Ours) | | 87.5 | 83.0 | 94.4 | 89.6 |
| IDS (Ours, Re-weighting Real Data) | | **87.9** | **83.4** | **94.9** | **90.1** |
| MAE (Xiao et al., 2023) | ViT-B/16 | 82.7 | 83.5 | 93.8 | 89.3 |
| MAE with Synthetic Data | | 83.5 | 82.7 | 94.0 | 89.0 |
| MW-Net (Shu et al., 2019) | | 83.9 | 82.7 | 93.8 | 89.3 |
| OTR (Guo et al., 2022) | | 85.5 | 81.6 | 93.2 | 89.3 |
| IE (Chhabra et al., 2024) | | 83.5 | 82.7 | 93.8 | 89.1 |
| CBF (He et al., 2023a) | | 84.6 | 81.8 | 93.8 | 89.2 |
| REVAR (Jain et al., 2024) | | 84.0 | 82.7 | 93.8 | 89.3 |
| IDS (Ours) | | 86.3 | 84.1 | 94.7 | 90.1 |
| IDS (Ours, Re-weighting Real Data) | | **86.8** | **84.8** | **95.5** | **90.7** |

Table 2: Performance comparisons between IDS models and SOTA baselines on COVIDx (in accuracy). Comparisons with more baselines are deferred to Table 9 in Section D.7 of the appendix.

| Method | Architecture | Covid-19 Sensitivity | Accuracy |
|---|---|---|---|
| MAE (Xiao et al., 2023) | | 94.5 | 95.2 |
| MAE with Synthetic Data | | 98.0 | 95.4 |
| MW-Net (Shu et al., 2019) | | 98.1 | 96.0 |
| OTR (Guo et al., 2022) | | 98.0 | 96.2 |
| IE (Chhabra et al., 2024) | ViT-S/16 | 98.0 | 96.0 |
| CBF (He et al., 2023a) | | 98.4 | 96.1 |
| REVAR (Jain et al., 2024) | | 98.2 | 96.2 |
| IDS (Ours) | | 98.8 | 97.1 |
| IDS (Ours, Re-weighting Real Data) | | 99.1 | 97.5 |
| MAE (Xiao et al., 2023) | | 95.5 | 95.3 |
| MAE with Synthetic Data | | 98.0 | 95.5 |
| MW-Net (Shu et al., 2019) | | 98.5 | 96.1 |
| OTR (Guo et al., 2022) | | 98.0 | 96.1 |
| IE (Chhabra et al., 2024) | ViT-B/16 | 98.0 | 96.0 |
| CBF (He et al., 2023a) | | 98.1 | 96.2 |
| REVAR (Jain et al., 2024) | | 98.2 | 96.3 |
| IDS (Ours) | | 99.0 | 97.3 |
| IDS (Ours, Re-weighting Real Data) | | 99.3 | 97.7 |

Table 3: Performance comparison between IDS models and SOTA baselines on NIH ChestX-ray14. Comparisons with more baselines are deferred to Table 10 in Section D.7 of the appendix.

| Method | Architecture | mAUC |
|---|---|---|
| MAE (Xiao et al., 2023) | | 82.3 |
| MAE with Synthetic Data | | 81.8 |
| MW-Net (Shu et al., 2019) | | 82.0 |
| OTR (Guo et al., 2022) | | 82.0 |
| IE (Chhabra et al., 2024) | ViT-S/16 | 82.1 |
| CBF (He et al., 2023a) | | 82.1 |
| REVAR (Jain et al., 2024) | | 82.1 |
| IDS (Ours) | | 82.7 |
| IDS (Ours, Re-weighting Real Data) | | 83.2 |
| MAE (Xiao et al., 2023) | | 83.0 |
| MAE with Synthetic Data | | 82.1 |
| MW-Net (Shu et al., 2019) | | 82.3 |
| OTR (Guo et al., 2022) | | 82.3 |
| IE (Chhabra et al., 2024) | ViT-B/16 | 82.5 |
| CBF (He et al., 2023a) | | 82.5 |
| REVAR (Jain et al., 2024) | | 82.5 |
| IDS (Ours) | | 83.4 |
| IDS (Ours, Re-weighting Real Data) | | 83.9 |

**COVIDx.** Table 2 compares the competing data selection and data re-weighting methods with our IDS for GDA on COVIDx. The base models, ViT-S and ViT-B, fine-tuned on the COVIDx dataset with synthetic data, achieve an accuracy of $95.4\%$ and $95.5\%$, respectively. Both IDS-ViT-S and IDS-ViT-B outperform their respective base models trained with synthetic data, with accuracy improvements of $1.7\%$ and $1.8\%$, respectively. IDS-ViT-B achieves a new state-of-the-art top-1 accuracy of $97.3\%$, with a $1.0\%$ improvement over the best competing baseline, REVAR, highlighting the efficacy of employing IDS for GDA on the COVIDx dataset. Moreover, applying IDS to re-weight both the real data and the synthetic data further boosts the performance of IDS. For example, IDS-ViT-B re-weighting both the synthetic data and the real data outperforms IDS-ViT-B re-weighting only the synthetic data by $0.4\%$ in mAUC, demonstrating the merits of IDS in selecting informative samples in both real data and synthetic data. Comparisons with additional baseline methods are provided in Table 9 in Section D.7 of the appendix.

**NIH ChestX-ray14.** Table 3 compares the competing data selection and data re-weighting methods with our IDS for GDA on the NIH ChestX-ray14 dataset. NIH ChestX-ray14 is an especially challenging dataset for GDA as it is a multi-label thorax disease classification dataset with 14 labels. All competing data selection methods and data re-weighting methods lead to even worse results than the baseline models trained without synthetic data. In contrast, IDS leads to improved performance over the baseline models and significantly outperforms competing data selection and data re-weighting methods. For instance, the base ViT-B network achieves a mean AUC (mAUC) of $83.0\%$, but the performance of ViT-B trained with synthetic data decreases to $82.1\%$. Although both data selection and data re-weighting methods bring improvements over the baseline trained with synthetic data, their performance remains worse than the baseline trained without synthetic data. In contrast, IDS-ViT-B outperforms the base model ViT-B trained without synthetic data by $0.4\%$, achieving an mAUC of $83.4\%$. IDS-ViT-B outperforms the best competing data re-weighting method, REVAR, by $0.9\%$ in mAUC. Moreover, applying IDS to re-weight both the real data and the synthetic data further boosts the performance. For example, IDS-ViT-B re-weighting both the synthetic data and the real data outperforms IDS-ViT-B re-weighting only the synthetic data by $0.5\%$ in mAUC. Comparisons with more baseline methods are available in Table 10 in Section D.7 of the appendix.

**Improvement Significance Analysis** To verify whether the improvements by our proposed IDS over existing methods are statistically significant and out of the range of error, we train both IDS and the leading baseline methods on different datasets from Table 1, Table 2, and Table 3 for 10 times with different seeds for random initialization of the networks and train/val/test splits. Subsequently, we perform the t-test between the results of IDS and the results of the best baseline methods on different datasets to assess if the improvement of IDS is statistically significant. The mean and standard deviation of the results and the p-values of the t-test are shown in Table 4 in Section D.3 of the appendix. The t-test results suggest that the improvements of IDS over the baseline methods is statistically significant with $p \ll 0.05$, and it is not caused by random error.

### 4.3 ABLATION STUDY

**Study on the Correlation between Disease Localization and Importance Weights.** In this section, we predict the disease localization areas using Grad-CAM heatmap (Selvaraju et al., 2017) and

assess the quality of synthetic images by computing IoU scores between the disease localization areas and the ground-truth bounding box of the disease. Following (Xiao et al., 2023), the predicated disease localization area is generated with the thresholded Grad-CAM heatmap. The threshold is set to 0.3 throughout all the experiments. A synthetic image with a higher IoU score between the disease localization area and the ground-truth bounding box is considered a more informative sample for the corresponding disease because a higher IoU means a larger overlap between the predicted disease localization area and the ground-truth bounding box of the disease. As illustrated in the examples in Figure 2, disease localization areas by IDS usually overlap more with the ground-truth bounding boxes than the competing baselines with higher IoU scores. To study whether more informative synthetic images receive higher importance weights by our IDS and the competing baselines, we analyze the correlation between the IoU scores and the importance weights predicted by IDS and baseline data re-weighting methods. Since the ground-truth disease bounding boxes for synthetic images are not available, we conduct the study on Cardiomegaly, which is a disease usually detected in a fixed region around the heart in the chest X-ray (Amin & Siddiqui, 2019). We use the ground-truth bounding box of Cardiomegaly from the test set of the NIH ChestX-rays14 (Wang et al., 2017) dataset as the ground-truth bounding box in our study.

The correlation between the individual IoU scores and importance weights is illustrated in the second row of Figure 1. Results on NIH-ChesX-ray14 are deferred to Figure 4 in Section D.2 of the appendix. Linear regression is performed between the individual IoU scores and importance weights to visualize the correlation. It is observed from the results that synthetic images with higher importance weights learned by IDS tend to have higher IoU scores, which suggests that our IDS renders higher importance weights for truly more informative synthetic images. In contrast, there is either no positive correlation, OTR (Guo et al., 2022), or only a tiny positive correlation, REVAR (Jain et al., 2024), between the importance weights of the IoU scores. We also apply the SCC to quantitatively measure the correlation between the individual VIB values and the importance weights of synthetic data. The SCC for IDS is 0.184, which is much higher than the SCC of 0.006 for the baseline method, REVAR. The SCC results demonstrate that the importance weights learned by IDS show much stronger positive correlations with the IoU scores compared to the baseline methods.

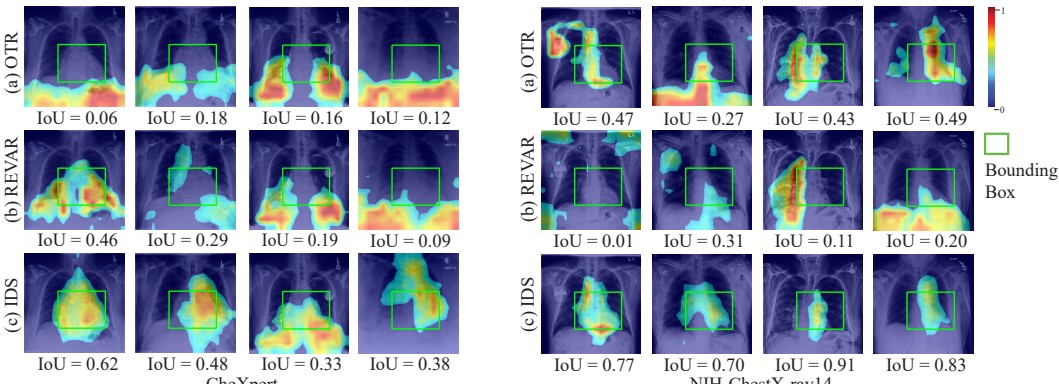

Figure 2: Grad-CAM visualization results on synthetic images for the disease Cardiomegaly from the CheXpert (left) and NIH ChestX-ray14 (right) datasets. The Grad-CAM visualizations are shown for (a) OTR, (b) REVAR, and (c) IDS in the first, second, and third rows, respectively. The green boxes represent the ground-truth bounding boxes. These visualizations illustrate that IDS consistently exhibits better disease localization ability compared to OTR (Guo et al., 2022) and REVAR (Jain et al., 2024), as reflected by the higher IoU scores.

## 5 CONCLUSION

In this paper, we propose Informative Data Selection (IDS), a novel method designed to re-weight synthetic images in Generative Data Augmentation (GDA) based on an information theoretic measure, the Information Bottleneck (IB). IDS trains a sample re-weighting network to minimize the IB loss on the synthetic data, such that the IB principle, learning features more correlated with the outputs and less correlated with the inputs, is better adhered. Extensive experiments and ablation studies demonstrate that IDS successfully assigns higher weights to more informative synthetic images for thorax disease classification and significantly outperforms existing data selection and data re-weighting methods for GDA.

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

## A  PROOF OF THEOREM 3.1

**Lemma A.1.**

$$I(\widehat{Z}, X) \leq \frac{1}{n} \sum_{i=1}^{n} \sum_{a=1}^{A} \sum_{b=1}^{B} \phi(\widehat{z}_j, a)\phi(x_i, b) \log \phi(x_i, b)$$

$$- \frac{1}{n^2} \sum_{i=1}^{n} \sum_{j=1}^{n} \sum_{b=1}^{B} \phi(x_i, b) \log \phi(X_j, b) \tag{8}$$

*Proof.* By the log sum inequality, we have

$$I(\widehat{Z}, X)$$

$$= \sum_{a=1}^{A} \sum_{b=1}^{B} \Pr\left[\widehat{Z} \in a, X \in b\right] \log \frac{\Pr\left[\widehat{Z} \in a, X \in b\right]}{\Pr\left[\widehat{Z} \in a\right] \Pr\left[X \in b\right]}$$

$$\leq \frac{1}{n^2} \sum_{i=1}^{n} \sum_{j=1}^{n} \sum_{a=1}^{A} \sum_{b=1}^{B} \phi(\widehat{z}_j, a)\phi(x_i, b) \left(\log\left(\phi(\widehat{z}_j, a)\phi(x_i, b)\right)\right.$$

$$- \log\left(\phi(\widehat{z}_j, a)\phi(X_j, b)\right)\big)$$

$$= \frac{1}{n^2} \sum_{i=1}^{n} \sum_{j=1}^{n} \sum_{a=1}^{A} \sum_{b=1}^{B} \phi(\widehat{z}_j, a)\phi(x_i, b) \log \phi(x_i, b)$$

$$- \frac{1}{n^2} \sum_{i=1}^{n} \sum_{j=1}^{n} \sum_{a=1}^{A} \sum_{b=1}^{B} \phi(\widehat{z}_j, a)\phi(x_i, b) \log \phi(X_j, b)$$

$$= \frac{1}{n} \sum_{i=1}^{n} \sum_{a=1}^{A} \sum_{b=1}^{B} \phi(\widehat{z}_j, a)\phi(x_i, b) \log \phi(x_i, b)$$

$$- \frac{1}{n^2} \sum_{i=1}^{n} \sum_{j=1}^{n} \sum_{a=1}^{A} \sum_{b=1}^{B} \phi(\widehat{z}_j, a)\phi(x_i, b) \log \phi(X_j, b). \tag{9}$$

$\square$

**Lemma A.2.**

$$I(\widehat{Z}, Y) \geq \frac{1}{n} \sum_{a=1}^{A} \sum_{y=1}^{C} \sum_{i=1}^{n} \phi(\widehat{z}_j, a) \mathbb{1}_{\{y_i = y\}} \log Q(\widehat{Z} \in a | Y = y) \tag{10}$$

*Proof.* Let $Q(\widehat{Z}|Y)$ be a variational distribution. We have

$$I(\widehat{Z}, Y)$$

$$= \sum_{a=1}^{A} \sum_{y=1}^{C} \Pr\left[\widehat{Z} \in a, Y = y\right] \log \frac{\Pr\left[\widehat{Z} \in a, Y = y\right]}{\Pr\left[\widehat{Z} \in a\right] \Pr\left[Y = y\right]}$$

$$= \sum_{a=1}^{A} \sum_{y=1}^{C} \Pr\left[\widehat{Z} \in a, Y = y\right] \log \frac{\Pr\left[\widehat{Z} \in a | Y = y\right] Q(\widehat{Z} \in a | Y = y)}{\Pr\left[\widehat{Z} \in a\right] Q(\widehat{Z} \in a | Y = y)}$$

$$\geq \sum_{a=1}^{A} \sum_{y=1}^{C} \Pr\left[\widehat{Z} \in a, Y = y\right] \log \frac{\Pr\left[\widehat{Z} \in a | Y = y\right]}{Q(\widehat{Z} \in a | Y = y)}$$

$$+ \sum_{a=1}^{A} \sum_{y=1}^{C} \Pr\left[\widehat{Z} \in a, Y = y\right] \log \frac{Q(\widehat{Z} \in a | Y = y)}{\Pr\left[\widehat{Z} \in a\right]}$$

$$= \text{KL}\left(P(\widehat{Z}|Y) \middle\| Q(\widehat{Z}|Y)\right)$$

$$+ \sum_{a=1}^{A} \sum_{y=1}^{C} \Pr\left[\widehat{Z} \in a, Y = y\right] \log \frac{Q(\widehat{Z} \in a | Y = y)}{\Pr\left[\widehat{Z} \in a\right]}$$

$$\geq \sum_{a=1}^{A} \sum_{y=1}^{C} \Pr\left[\widehat{Z} \in a, Y = y\right] \log \frac{Q(\widehat{Z} \in a | Y = y)}{\Pr\left[\widehat{Z} \in a\right]}$$

$$= \sum_{a=1}^{A} \sum_{y=1}^{C} \Pr\left[\widehat{Z} \in a, Y = y\right] \log Q(\widehat{Z} \in a | Y = y) + H\left(P(\widehat{Z})\right)$$

$$\geq \sum_{a=1}^{A} \sum_{y=1}^{C} \Pr\left[\widehat{Z} \in a, Y = y\right] \log Q(\widehat{Z} \in a | Y = y)$$

$$\geq \frac{1}{n} \sum_{a=1}^{A} \sum_{y=1}^{C} \sum_{i=1}^{n} \phi(\widehat{z}_j, a) \mathbb{1}_{\{y_i = y\}} \log Q(\widehat{Z} \in a | Y = y). \tag{11}$$

$$\square$$

# B   INFORMATION ON DIFFUSION MODELS

## B.1   FORMULATIONS OF DIFFUSION MODELS

**Diffusion models (DMs)** are latent variable models that conceptualize data $x^0$ as a Markov chain progressing from $x_T$ to $x^0$, with all intermediate variables maintaining consistent dimensions. These models involve two primary Markovian processes: a forward diffusion process defined as $q(x^{(1:T)} \mid x^0) = \prod_{t=1}^{T} q(x^{(t)} \mid x^{(t-1)})$ and a reverse denoising process described by $p_\omega(x_{0:T}) = p(x_T) \prod_{t=1}^{T} p_\omega(x^{(t-1)} \mid x^{(t)})$. The forward process methodically incorporates Gaussian noise into data $x^{(t)}$:

$$q(x^{(t)} \mid x^{(t-1)}) = \mathcal{N}(x^{(t)}; \sqrt{1 - \beta^{(t)}} x^{(t-1)}, \beta^{(t)} \mathbf{I}), \tag{12}$$

where the hyperparameter series $\beta^{(1:T)}$ dictates the noise level added at each step $t$. The chosen $\beta^{(1:T)}$ ensures that samples $x_T$ approximate standard Gaussian distributions, i.e., $q(x_T) \approx \mathcal{N}(0, \mathbf{I})$. Typically, this forward process $q$ is not adjustable post-definition.

The generation method for DMs involves learning a parameter-driven reverse denoising process to systematically purify the noisy variables $x_{T:1}$ back to the pristine data $x^0$:

$$p_\omega(x^{(t-1)} \mid x^{(t)}) = \mathcal{N}(x^{(t-1)}; \mu_\omega(x^{(t)}, t), (\rho^{(t)})^2 \mathbf{I}), \tag{13}$$

with the initial distribution $p(x_T)$ set as $\mathcal{N}(0, \mathbf{I})$. The model utilizes neural networks like U-Nets or Transformers for calculating means $\mu_\omega$, with variances $\rho^{(t)}$ usually predefined.

In terms of optimization, the forward process $q(x^{(1:T)}|x^0)$ is treated as a fixed posterior, against which the reverse process $p_\omega(x_{0:T})$ is trained to enhance the variational lower bound of the data likelihood. Direct likelihood optimization can lead to significant training instability. An alternative simple surrogate objective suggested is:

$$\mathcal{L}_{\text{DM}} = \mathbb{E}_{x^0, \varepsilon \sim \mathcal{N}(0, \mathbf{I}), t} \left\| \varepsilon - \varepsilon_\omega(x^{(t)}, t) \right\|_2^2, \tag{14}$$

where the model $\varepsilon_\omega$ predicts the noise vector $\varepsilon$ to clarify diffused samples $x^{(t)}$ at every stage $t$ back to $x^{(t-1)}$. Post-training, samples are generated through iterative ancestral sampling:

$$x^{(t-1)} = \frac{1}{\sqrt{1 - \beta^{(t)}}} \left( x^{(t)} - \frac{\beta^{(t)}}{\sqrt{1 - (\alpha^{(t)})^2}} \varepsilon_\omega(x^{(t)}, t) \right) + \rho^{(t)} \varepsilon, \tag{15}$$

starting from a Gaussian prior $x_T \sim p(x_T) = \mathcal{N}(x_T; 0, \mathbf{I})$.

**Latent Diffusion Models (LDMs)** enhance standard Diffusion Models by introducing a latent space that reduces the dimensionality of the data involved in the diffusion process. Initially, data $x^0$ is encoded to a lower-dimensional latent form $h^0$. The forward process in LDMs involves:

$$q(h^{(t)} \mid h^{(t-1)}) = \mathcal{N}(h^{(t)}; \sqrt{1 - \beta^{(t)}} h^{(t-1)}, \beta^{(t)} I), \tag{16}$$

and the reverse process reconstructs the original clean latent state $h^0$ from $h_T$ by:

$$p_\omega(h^{(t-1)} \mid h^{(t)}) = \mathcal{N}(h^{(t-1)}; \mu_\omega(h^{(t)}, t), (\rho^{(t)})^2 I), \tag{17}$$

followed by transforming the reconstructed latent data $h^0$ back to the original data space. The training loss for LDM is

$$\mathcal{L}_{\text{LDM}} = \mathbb{E}_{h_e(x), \epsilon \sim \mathcal{N}(0, I), t} \left\| \epsilon - \epsilon_\omega(h^{(t)}, t, y) \right\|_2^2, \tag{18}$$

**Classifier-Free Guidance (CFG)** merges a conditional and an unconditional noise predictor in the sampling process to elevate sample quality and provide class guidance. This technique can be seamlessly integrated into LDMs, formulated as:

$$h^{(t-1)} = \frac{1}{\sqrt{1 - \beta^{(t)}}} \left( h^{(t)} - \frac{\beta^{(t)}}{\sqrt{1 - (\alpha^{(t)})^2}} \tilde{\varepsilon}^{(t)} \right) + \rho^{(t)} \varepsilon, \tag{19}$$

where $\tilde{\varepsilon}^{(t)} = (1 + \omega) \varepsilon_\omega(h^{(t)}, y, t) - \gamma \varepsilon_\omega(h^{(t)}, t)$, and $\gamma$ is the guidance factor, optimizing the sampling process for specific outcomes.

Algorithm 1 describes the training algorithm of the LDM. Algorithm 2 describes the generation process of the synthetic training set.

---

**Algorithm 1** Training Algorithm of LDM

**Input:** The original training set $\mathcal{D}_{\text{real}} = \{x_i, y_i\}_{i=1}^N$, the encoder $v_e$ of the fixed pre-trained VAE, and the training epochs of the LDM $t_{\text{LDM}}$.
**Output:** The parameters of the LDM $\omega$.
1: Initialize the parameter $\omega$ of the LDM.
2: Encode input features $\{x_i\}_{i=1}^N$ to the latent features $\{h_i\}_{i=1}^N$ using the encoder $v_e$ such that $h_i = v_e(x_i)$.

3: **for** $t = 1, 2, \ldots, t_{\text{LDM}}$ **do**
4:     Update $\omega$ by mini-batch SGD on $\{h_i\}_{i=1}^N$ using the loss $\mathcal{L}_{\text{LDM}}$ in Equation (18).
5: **end for**
6: **return** The parameters of the LDM $\omega$.

---

**Algorithm 2** Generation of Synthetic Training Set

**Input:** The labels of the synthetic training set $\{\widehat{y}_j\}_{j=1}^M$, the parameters of the LDM $\omega$, and the decoder $v_d$ of the fixed pre-trained VAE.
**Output:** The synthetic training set $\mathcal{D}_{\text{syn}} = \{\widehat{x}_i, \widehat{y}_i\}_{j=1}^M$.
1: **for** $j = 1, 2, \ldots, M$ **do**
2:     Sample a Gaussian noise $\epsilon \sim \mathcal{N}(0, I)$
3:     Generate synthetic latent feature $\widehat{h}_j$ from $\epsilon$ with the LDM using Equation (17) in Section B of the appendix.
4:     Decode latent feature $\widehat{h}_j$ to the synthetic input feature $\widehat{x}_j$ by $\widehat{x}_j = v_d(\widehat{h}_j)$.
5: **end for**
6: **return** The synthetic training set $\mathcal{D}_{\text{syn}} = \{\widehat{x}_i, \widehat{y}_i\}_{j=1}^M$.

---

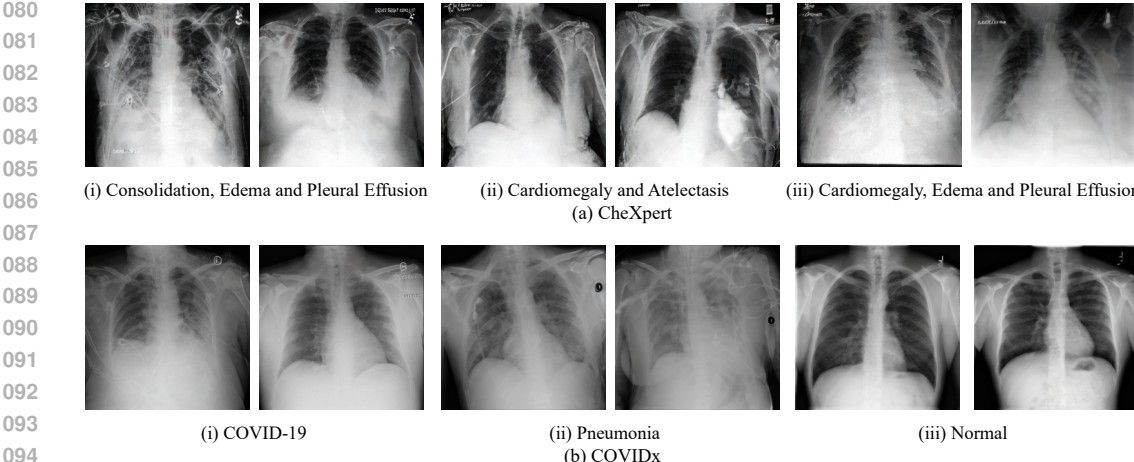

(i) Consolidation, Edema and Pleural Effusion    (ii) Cardiomegaly and Atelectasis    (iii) Cardiomegaly, Edema and Pleural Effusion

(a) CheXpert

(i) COVID-19    (ii) Pneumonia    (iii) Normal

(b) COVIDx

Figure 3: Examples of synthetic images generated using a diffusion model trained on the (a) CheXpert and (b) COVIDx datasets, displayed in the first and second rows, respectively. In the first row (CheXpert), the images depict the following medical conditions: (i) Consolidation, Edema, and Pleural Effusion; (ii) Cardiomegaly and Atelectasis; (iii) Cardiomegaly and Pleural Effusion. In the second row (COVIDx), the images correspond to: (i) COVID-19; (ii) Pneumonia; and (iii) Normal (no disease).

### B.2 DATA GENERATION WITH THE DIFFUSION MODELS

We train the Diffusion Transformer (DiT) on $256 \times 256$ images, following the protocol outlined in (Peebles & Xie, 2023). The training process spans 2,800 epochs with a global batch size of 512, distributed across four NVIDIA A100 GPUs. A constant learning rate of $1 \times 10^{-4}$ is maintained throughout the training. After training, we generate synthetic images using a classifier-free guidance (CFG) scale of 4.0 with 128 sampling steps. The synthetic dataset is constructed to mirror the label distribution of the real data, ensuring that disease co-occurrence patterns are preserved. Figure 3 presents examples of synthetic images generated by the diffusion model for various thorax diseases. We then integrate these synthetic images into the training sets for COVIDx, CheXpert and NIH-ChestX-ray14. Specifically, we augment the CheXpert, COVIDx and NIH-ChestX-ray14 training sets with $1.0n$ synthetic images, where '$n$' denotes the number of images in the official training split of each respective dataset. To ensure fair comparison, all the other baselines are augmented with a similar number of synthetic images.

### B.3 COMPUTATION OF $Q^{(t)}(\tilde{\mathbf{x}}|Y)$

The variational distribution $Q^{(t)}(\tilde{\mathbf{x}}|Y)$ can be computed by

$$Q^{(t)}(\widehat{Z} \in a|Y = y) = \Pr\left[\widehat{Z} \in a|Y = y\right]$$

$$= \frac{\sum\limits_{i=1}^{n} \phi(\widehat{z}_j, a)\,\mathbb{I}_{\{y_i = y\}}}{\sum\limits_{i=1}^{n} \mathbb{I}_{\{y_i = y\}}}. \tag{20}$$

## C ALGORITHM OF IDS

The algorithm for the training process of IDS is described in Algorithm 3.

---

**Algorithm 3** Algorithm of IDS

---

**Input:** The augmented training set $\mathcal{D}_{\text{aug}}$, the synthetic training set $\mathcal{D}_{\text{syn}}$, the original training set $\mathcal{D}_{\text{real}}$, epoch number $t_{\max}$.

1: Initialize the classifier network parameters $\Theta^{(0)}$ and the sample re-weighting network parameters $\theta^{(0)}$.
2: **for** $t = 1, 2, \ldots, t_{\max}$ **do**
3:    Compute the class centroids of the input features and image representations $\mathcal{C}(\theta, \Theta^{(t-1)})$.
4:    Update $\theta^{(t)}$ by applying mini-batch gradient descent on $\mathcal{D}_{\text{syn}}$ following Equation (6).
5:    Update $\Theta^{(t)}$ byapplying mini-batch gradient descent on $\mathcal{D}_{\text{aug}}$ following Equation (7).
6:    Compute $Q^{(t)}(\widehat{Z} \in a | \widehat{Y} = y)$ by Eq. (20) in the supplementary.
7: **end for**
8: **return** The trained weights $\Theta$ of the classifier network $f_{\Theta}(\cdot)$ and the trained weights $\theta$ of the sample re-weighting network $g_{\theta}(\cdot)$.

---

# D ADDITIONAL EXPERIMENTS

## D.1 ADDITIONAL IMPLEMENTATION DETAILS AND EXPERIMENTAL SETUPS

The fine-tuning process is performed for 75 epochs with the ADAM optimizer and a batch size of 1024. A cosine decay schedule is used. The initial learning rate $\mu$ is determined through cross-validation for each model and dataset. The weight decay is set to 0.05, and the momentum parameters $\beta_1$ and $\beta_2$ are set to 0.9 and 0.999 for all the experiments. We compare our IDS models with several data selection and sample reweighting methods, including Influence Estimation (Chhabra et al., 2024), Classifier-based Filtering (CBF) (He et al., 2023a), MW-Net (Shu et al., 2019), OTR (Guo et al., 2022), and REVAR (Jain et al., 2024). To ensure a fair comparison, all baseline models undergo an additional 75 epochs of fine-tuning. The mean Area Under the Curve (mAUC) is used as the metric for the multi-label disease classification datasets CheXpert and NIH ChestX-ray14. Accuracy is used as the metric for the single-label disease classification dataset COVIDx.

**CheXpert.** The CheXpert dataset (Irvin et al., 2019) consists of $224, 316$ chest X-ray images from $65, 240$ patients, with $191, 028$ images used for training. Each X-ray is labeled with radiology reports indicating the presence of 14 thoracic diseases. To measure the effectiveness of our approach, we compute the mean Area Under the Curve (AUC) across five selected disease categories and compare our results against state-of-the-art baseline models.

**COVIDx.** The COVIDx dataset (Version 9A) (Pavlova et al., 2022) comprises 30,386 chest X-ray images from $17, 026$ unique patients. Following the partitioning strategy used in previous studies (Pavlova et al., 2022; Xiao et al., 2023), the dataset is divided into $29, 986$ images for training across four classes, and 400 images for testing, categorized into three classes. For objective evaluation and consistency with prior methodologies, we report the Top-1 accuracy on the test set, which contains three classes.

**NIH ChestX-ray14.** NIH ChestX-ray14 (Wang et al., 2017) is a large-scale dataset comprising $112, 120$ chest X-ray images collected from $30, 805$ unique patients. Each image may have multiple labels from 14 disease categories, allowing for multi-label classification tasks. Following the official data split provided by Wang et al. (2017), we use $75, 312$ images for training and $25, 596$ images for testing. The raw images have a resolution of $1024 \times 1024$ pixels. In our experiments, we resize the images to $224 \times 224$ pixels to match the input requirements of our models. We report the mean Area Under the Curve (AUC) across all 14 disease classes and conduct a comprehensive comparison with 18 widely recognized and influential baseline methods.

## D.2 ADDITIONAL STUDY ON THE CORRELATION BETWEEN DISEASE LOCALIZATION AND IMPORTANCE WEIGHTS

Figure 4 illustrates the correlation analysis between IoU scores for disease localization and importance weights on Cardiomegaly for OTR (Guo et al., 2022), REVAR (Jain et al., 2024) and IDS in the NIH-ChestX-ray14 dataset.

As illustrated in Figure 2, the disease localization areas predicted by IDS tend to overlap more with the ground-truth bounding boxes than those predicted by competing baselines, yielding higher IoU scores. To investigate whether IDS assigns higher importance weights to more informative synthetic

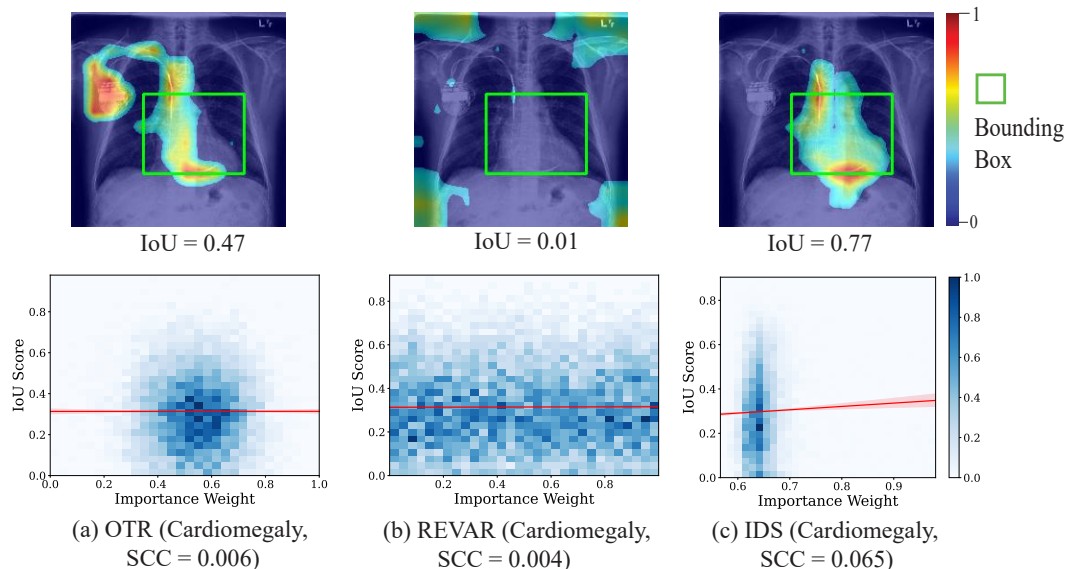

Figure 4: Figures in the first row are examples of thresholded Grad-CAM visualization for OTR, REVAR, and IDS. For each of the examples, we also present the ground-truth bounding box for the disease Cardiomegaly. The thresholded heatmap areas are considered as the disease localization areas. IoU score between the disease localization area and the ground-truth bounding box is shown below each example. A synthetic image with a higher IoU score is considered a more informative sample for this disease as a larger portion of the predicted disease localization area overlaps with the ground-truth bounding box of the disease. Figures in the second row illustrate the correlation between IoU scores for disease localization and importance weights on Cardiomegaly for OTR (Guo et al., 2022), REVAR (Jain et al., 2024) and IDS in the NIH-ChestX-ray14 dataset. The disease name and Spearman Correlation Coefficients (SCC) (Spearman, 1961) are attached in the parenthesis. A larger absolute value of a positive SCC between two variables indicates a stronger positive correlation, which refers to a correlation between two variables where as one variable increases, the other variable tends to increase as well. The range of IoU and the range of the importance weight, which is $[0, 1] \times [0, 1]$, is divided into $30 \times 30$ cells evenly, and the color of each cell is proportional to the number of synthetic images whose IoU sores and importance weights fall in that cell. As a result, a cell with more blue indicates more synthetic images falling in that cell. The red lines in the figures are the linear regression results between the IoU scores and the importance weights, which visualizes the correlation. It can be observed that the linear regressors in red suggest a stronger positive correlation between the IoU scores and the importance weights by our IDS than that for competing baselines, which is further quantitatively evidenced by the higher SCC for IDS than the competing baselines.

images, we analyze the correlation between IoU scores and importance weights predicted by IDS and other baseline data re-weighting methods. The second row of Figure 4 illustrates the correlation between individual IoU scores and importance weights. Linear regression is performed to visualize this relationship. The results show that synthetic images assigned higher importance weights by IDS generally have higher IoU scores, indicating that IDS effectively identifies and prioritizes more informative synthetic images. In contrast, there is only a weak positive correlation between importance weights and IoU scores for OTR (Guo et al., 2022) and REVAR (Jain et al., 2024). To further quantify this correlation, we apply the Spearman Correlation Coefficient (SCC) (Spearman, 1961). The SCC for IDS is 0.065, significantly higher than the SCC of 0.004 for REVAR, demonstrating that IDS assigns importance weights that are more strongly correlated with IoU scores compared to baseline methods.

### D.3 IMPROVEMENT SIGNIFICANCE ANALYSIS

To verify that the improvement of our proposed IDS on existing methods is statistically significant and out of the range of error, we train both IDS and the best baseline methods on different datasets from Table 1, Table 2, and Table 3 for 10 times with different seeds for random initialization of the networks and train/val/test splits. Next, we perform the t-test between the results of IDS and the results of the best baseline methods on different datasets to assess if the improvement of IDS is statistically significant. The mean and standard deviation of the results and the p-values of the t-test are shown in Table 4. It is observed that the largest p-value is $1.44 \times 10^{-10}$, which is less than 0.05. The t-test results suggest that the improvement of IDS over the baseline methods is statistically significant with $p \ll 0.05$, and it is not caused by random error.

Table 4: P-values of t-test between IDS the baseline methods with the best performance on CheXpert, COVIDx, and NIH ChestX-ray14.

| Dataset | Architecture | CheXpert (mAUC) | COVIDx (Accuracy) | NIH ChestX-ray14 (mAUC) |
|---|---|---|---|---|
| Best Baseline | ViT-S/16 | $89.2 \pm 0.067$ | $96.2 \pm 0.122$ | $82.3 \pm 0.045$ |
| IDS | | $89.6 \pm 0.112$ | $97.1 \pm 0.125$ | $82.7 \pm 0.052$ |
| p-value | - | $1.44 \times 10^{-10}$ | $3.20 \times 10^{-12}$ | $4.07 \times 10^{-13}$ |
| Best Baseline | ViT-B/16 | $89.3 \pm 0.045$ | $96.3 \pm 0.158$ | $83.0 \pm 0.051$ |
| IDS | | $90.1 \pm 0.096$ | $97.3 \pm 0.136$ | $83.4 \pm 0.065$ |
| p-value | - | $1.44 \times 10^{-15}$ | $1.44 \times 10^{-11}$ | $1.44 \times 10^{-12}$ |

### D.4 ABLATION STUDY AND TRAINING TIME ANALYSIS OF THE IDS

To evaluate the effectiveness and efficiency of different components in the IDS. We compare the disease classification performance and the training time of the baseline model ViT-B, the IDS model IDS-ViT-B, and two ablation models, which are IDS-ViT-B without VIB and IDS-ViT-B without the re-weighting network. The comparison is performed on the COVIDx dataset. The training time is evaluated on four NVIDIA A100 GPUs. The results are shown in Table 5. With only a 30% increase in the training time, IDS-ViT-B improves the classification accuracy on COVIDx by 2.0%, demonstrating the effectiveness of integrating these components into the baseline model. The ablation studies further confirm the individual contributions of the VIB and the re-weighting network, underlining the importance of both components in enhancing model performance while maintaining a manageable increase in computational demand.

Table 5: Ablation study of IDS with training time analysis. The training time is evaluated on four NVIDIA A100 GPUs.

| Methods | COVIDx (Accuracy) | Training Time (minutes/epoch) |
|---|---|---|
| ViT-B | 95.3 | 2.6 |
| IDS-ViT-B w/o VIB | 96.4 | 3.2 |
| IDS-ViT-B w/o Re-weighting Network | 96.7 | 2.8 |
| IDS-ViT-B | **97.3** | 3.4 |

### D.5 STUDY ON THE DIFFUSION MODELS FOR THE DATA GENERATION IN THE IDS

To evaluate the impact of the diffusion model used for the data generation in the IDS, we compare the performance of IDS-ViT-B using three different diffusion models for data generation, which are DiT-B, DiT-L, and DiT-XL Peebles & Xie (2023). The data generation time and the classification accuracy on the COVIDx dataset are shown in Table 6. It is observed that the performance of the IDS model is not sensitive to the selection of the diffusion models used for data generation. The IDS-ViT-B based on the largest DiT model DiT-XL only outperforms the IDS-ViT-B based on the smallest DiT model DiT-B by 0.2% in classification accuracy on COVIDx, demonstrating the merit of IDS in mitigating the noise in the synthetic data generated by diffusion models. In addition, the results in Table 6 show that the synthetic data generation process with the diffusion models in IDS is efficient, with less than 0.01 seconds/image.

Table 6: Performance comparison of IDS-ViT-B utilizing different diffusion models for data generation. The data generation time is evaluated on four NVIDIA A100 GPUs.

| Methods | COVIDx (Accuracy) | Generation Time (seconds/image) |
|---|---|---|
| ViT-B | 95.3 | - |
| IDS-ViT-B (DiT-B) | 97.1 | 0.095 |
| IDS-ViT-B (DiT-L) | **97.3** | 0.151 |
| IDS-ViT-B (DiT-XL) | **97.3** | 0.176 |

## D.6  COMPARISON BETWEEN IDS AND ACTIVE LEARNING METHODS

Active learning (AL) methods aim to minimize the effort required for labeling training data by strategically choosing the most informative instances for annotation (Sinha et al., 2019; Yoo & Kweon, 2019; Gao et al., 2020; Kushnir & Venturi, 2023; Yang et al., 2023; Chhabra et al., 2024). The selection of the data for annotation by active learning methods is usually achieved by identifying the most informative data points. Such a process works similarly to the data r-weighting process in IDS for identifying the most informative synthetic data. To show the advantage of IDS over active learning methods in selecting the most informative synthetic data, we compare IDS with two state-of-the-art active learning methods, which are CAMPAL (Yang et al., 2023) and SAAL (Chhabra et al., 2024). Both CAMPAL and SAAL are adopted to select data from the synthetic dataset generated by the diffusion models. The results are shown in Table 7. It is observed that IDS outperforms the competing active learning methods on all the datasets, demonstrating the superiority of IDS in selecting informative training samples compared to active learning methods.

Table 7: Comparison between IDS and active learning methods.

| Methods | COVIDx (mAUC) | Covid-19 (Accuracy) | NIH ChestX-ray14 (mAUC) |
|---|---|---|---|
| ViT-B | 89.3 | 95.3 | 83.0 |
| CAMPAL-ViT-B | 89.4 | 96.2 | 83.0 |
| SAAL-ViT-B | 89.3 | 95.9 | 83.1 |
| IDS-ViT-B | **89.6** | **97.3** | **83.4** |

## D.7  COMPARISON WITH MORE EXISTING WORKS ON THORAX DISEASE CLASSIFICATION

We compare our IDS models with more baselines for thorax disease classification on CheXpert, COVIDx, and NIH-ChestXray-14 in Table 8, Table 9, and Table 10, respectively.

**CheXpert.** Table 8 presents a performance comparison between additional baseline models and those enhanced by our Informative Data Selection (IDS) technique. For instance, IDS-ViT-B achieves significant improvements, with gains of up to 7.3% in mAUC over the baseline models. In addition to the overall mAUC, Table 8 also provides AUC scores for key thoracic diseases, including Atelectasis, Cardiomegaly, and Edema. These individual disease-specific results further emphasize the effectiveness of IDS, as it consistently boosts performance across various conditions. These findings highlight the superior capabilities of IDS-enhanced models compared to standard baselines on the CheXpert dataset.

**COVIDx.** Table 9 presents performance comparisons between additional baseline models and our IDS-enhanced models on the COVIDx dataset. For instance, IDS-ViT-B significantly outperforms the baseline models, with accuracy gains of up to 4.7%. Moreover, IDS-ViT-S and IDS-ViT-B achieve a state-of-the-art COVID-19 sensitivity of 99.0%, surpassing previous baselines by up to 11.9%. These results demonstrate the effectiveness of integrating IDS into transformer-based models for medical image analysis on the COVIDx dataset.

**NIH-ChestX-ray14.** Table 10 compares the performance of various state-of-the-art (SOTA) CNN-based and transformer-based models, including those enhanced by our Informative Data Selection (IDS) technique, on the NIH ChestX-ray14 dataset. The table includes models pre-trained on both ImageNet and X-rays. IDS-ViT-B shows significant improvements, achieving gains of up to 8.9% in mAUC and 8.2% for IDS-ViT-S over baseline models. These gains highlight the effectiveness of IDS in improving performance for thoracic disease classification. Furthermore, Table 10 presents mAUC scores for all methods, demonstrating that IDS-enhanced models consistently outperform other baseline methods, including both CNN and transformer-based architectures, on the NIH ChestX-ray14 dataset. These findings underscore the superior capabilities of IDS-enhanced models in addressing the challenges of thoracic disease classification.

## D.8  GRAD-CAM VISUALIZATION RESULTS ON NIH-CHESTX-RAY14

In this section, we present Grad-CAM visualization results on the NIH ChestX-ray14 dataset, which includes various disease labels such as Pneumothorax, Atelectasis, Mass, Cardiomegaly, Pneumonia, and Effusion. The dataset provides bounding box annotations for certain disease labels, which we use in our evaluations to assess the accuracy of localization. We visualize the regions in the input images that are responsible for the model's predictions on the ground-truth disease labels, comparing the performance of IDS against several baseline models, including MAE (Xiao et al., 2023),

Table 8: The performance of various state-of-the-art (SOTA) baseline methods on CheXpert. DN represents DenseNet, where the second best performance is underlined.

| Method | Architecture | Atelectasis | Cardiomegaly | Edema | mAUC (%) |
|---|---|---|---|---|---|
| Allaouzi et al.(Allaouzi & Ahmed, 2019) | | 72.0 | **88.0** | 87.0 | 82.8 |
| Irvin et al.(Irvin et al., 2019) | | 81.8 | 82.8 | **93.4** | 88.9 |
| Chexclusion (Seyyed-Kalantari et al., 2020) | | 81.2 | 83.0 | 88.3 | 87.3 |
| Pham et al.(Pham et al., 2021) | | **82.5** | 85.5 | 93.0 | 89.4 |
| BMTL (Hosseinzadeh Taher et al., 2021) | DN121 | - | - | - | 87.1 |
| DiRA (Haghighi et al., 2022) | | - | - | - | 87.6 |
| Label-assemble (Kang et al., 2021) | | 82.1 | 85.9 | 89.2 | 89.0 |
| MoCo v2 (Xiao et al., 2023) | | 78.5 | 77.9 | 92.8 | 88.7 |
| MAE (Xiao et al., 2023) | | 81.5 | 77.6 | 92.3 | 88.7 |
| MAE (Xiao et al., 2023) | | 83.5 | 81.8 | 94.0 | 89.2 |
| MAE with Synthetic Data | | 83.0 | 81.5 | 94.0 | 88.6 |
| MW-Net (Shu et al., 2019) | | 81.7 | 82.7 | 94.1 | 88.9 |
| OTR (Guo et al., 2022) | ViT-S/16 | 84.6 | 81.2 | 94.2 | 89.0 |
| IE (Chhabra et al., 2024) | | 81.7 | 82.0 | 94.2 | 88.9 |
| CBF (He et al., 2023a) | | 81.4 | 82.7 | 94.2 | 88.8 |
| REVAR (Jain et al., 2024) | | 83.0 | 82.7 | 94.0 | 89.0 |
| IDS (Ours) | | **87.5** | **83.0** | **94.4** | **89.6** |
| MAE (Xiao et al., 2023) | | 82.7 | 83.5 | 93.8 | 89.3 |
| MAE with Synthetic Data | | 83.5 | 82.7 | 94.0 | 89.0 |
| MW-Net (Shu et al., 2019) | | 83.9 | 82.7 | 93.8 | 89.3 |
| OTR (Guo et al., 2022) | ViT-B/16 | 85.5 | 81.6 | 93.2 | 89.3 |
| IE (Chhabra et al., 2024) | | 83.5 | 82.7 | 93.8 | 89.1 |
| CBF (He et al., 2023a) | | 84.6 | 81.8 | 93.8 | 89.2 |
| REVAR (Jain et al., 2024) | | 84.0 | 82.7 | 93.8 | 89.3 |
| IDS (Ours) | | **86.3** | **84.1** | **94.7** | **90.1** |

Table 9: Performance comparisons between IDS models and SOTA baselines on COVIDx (in accuracy). DN represents DenseNet.

| Method | Architecture | Covid-19 Sensitivity | Accuracy |
|---|---|---|---|
| COVIDNet-CXR Small (Wang et al., 2020) | - | 87.1 | 92.6 |
| COVIDNet-CXR Large (Wang et al., 2020) | - | 96.8 | 94.4 |
| MoCo v2 (Xiao et al., 2023) | DN121 | 94.5 | 94.0 |
| MAE (Xiao et al., 2023) | DN121 | 97.0 | 93.5 |
| MAE (Xiao et al., 2023) | | 94.5 | 95.2 |
| MAE with Synthetic Data | | 98.0 | 95.4 |
| MW-Net (Shu et al., 2019) | | 98.1 | 96.0 |
| OTR (Guo et al., 2022) | ViT-S/16 | 98.0 | 96.2 |
| IE (Chhabra et al., 2024) | | 98.0 | 96.0 |
| CBF (He et al., 2023a) | | 98.4 | 96.1 |
| REVAR (Jain et al., 2024) | | 98.2 | 96.2 |
| IDS (Ours) | | **98.8** | **97.1** |
| MAE (Xiao et al., 2023) | | 95.5 | 95.3 |
| MAE with Synthetic Data | | 98.0 | 95.5 |
| MW-Net (Shu et al., 2019) | | 98.5 | 96.1 |
| OTR (Guo et al., 2022) | ViT-B/16 | 98.0 | 96.1 |
| IE (Chhabra et al., 2024) | | 98.0 | 96.0 |
| CBF (He et al., 2023a) | | 98.1 | 96.2 |
| REVAR (Jain et al., 2024) | | 98.2 | 96.3 |
| IDS (Ours) | | **99.0** | **97.3** |

Table 10: Performance comparison of various state-of-the-art (SOTA) CNN-based and Transformer-based methods on NIH ChestX-ray14. RN, DN, and SwinT represent ResNet, DenseNet, and Swin Transformer.

| Method | Architecture | Pre-training | mAUC |
|---|---|---|---|
| Wang et al.(Wang et al., 2017) | RN50 | | 74.5 |
| Li et al.(Li et al., 2018) | RN50 | | 75.5 |
| LSE-LBA(Yao et al., 2018) | RN&DN | | 76.1 |
| Thorax-Net(Wang et al., 2019) | R152 | | 78.8 |
| MA(Ma et al., 2019) | R101 | | 79.4 |
| AGCL(Tang et al., 2018) | RN50 | | 80.3 |
| Baltruschat et al.(Baltruschat et al., 2019) | RN50 | | 80.6 |
| DNetLoc (Guendel et al., 2018) | DN121 | | 80.7 |
| CRAL(Guan & Huang, 2018) | DN121 | | 81.6 |
| Seyyed et al.(Seyyed-Kalantari et al., 2020) | DN121 | ImageNet-1K | 81.2 |
| CAN(Ma et al., 2020) | DN121(×2) | | 81.7 |
| Hermoza et al.(Hermoza et al., 2020) | DN121 | | 82.1 |
| XProtoNet(Kim et al., 2021) | DN121 | | 82.2 |
| DiRA(Haghighi et al., 2022) | DN121 | | 81.7 |
| ACPL (Liu et al., 2022) | DN121 | | 81.8 |
| SwinCheX (Taslimi et al., 2022) | SwinT | | 81.0 |
| Categorization (Xiao et al., 2023) | RN50 | | 81.8 |
| Categorization (Xiao et al., 2023) | DN121 | | 82.0 |
| MoCo v2 (Xiao et al., 2023) | DN121 | X-rays (0.3M) | 80.6 |
| MAE (Xiao et al., 2023) | DN121 | | 81.2 |
| MAE (Xiao et al., 2023) | | | 82.3 |
| MAE with Synthetic Data | | | 81.8 |
| MW-Net (Shu et al., 2019) | | | 82.0 |
| OTR (Guo et al., 2022) | ViT-S/16 | X-rays (0.3M) | 82.0 |
| IE (Chhabra et al., 2024) | | | 82.1 |
| CBF (He et al., 2023a) | | | 82.1 |
| REVAR (Jain et al., 2024) | | | 82.1 |
| IDS (Ours) | | | 82.7 |
| MAE (Xiao et al., 2023) | | | 83.0 |
| MAE with Synthetic Data | | | 82.1 |
| MW-Net (Shu et al., 2019) | | | 82.3 |
| OTR (Guo et al., 2022) | ViT-B/16 | X-rays (0.5M) | 82.3 |
| IE (Chhabra et al., 2024) | | | 82.5 |
| CBF (He et al., 2023a) | | | 82.5 |
| REVAR (Jain et al., 2024) | | | 82.5 |
| IDS (Ours) | | | **83.4** |

OTR (Guo et al., 2022), and REVAR (Jain et al., 2024). The visualizations in Figure 5 demonstrate that IDS tends to focus more accurately on areas inside the bounding boxes provided by the NIH ChestX-ray14 dataset, which correspond to the labeled disease regions. In contrast, the baseline models often activate regions outside the bounding boxes or irrelevant background areas, indicating less precise localization.

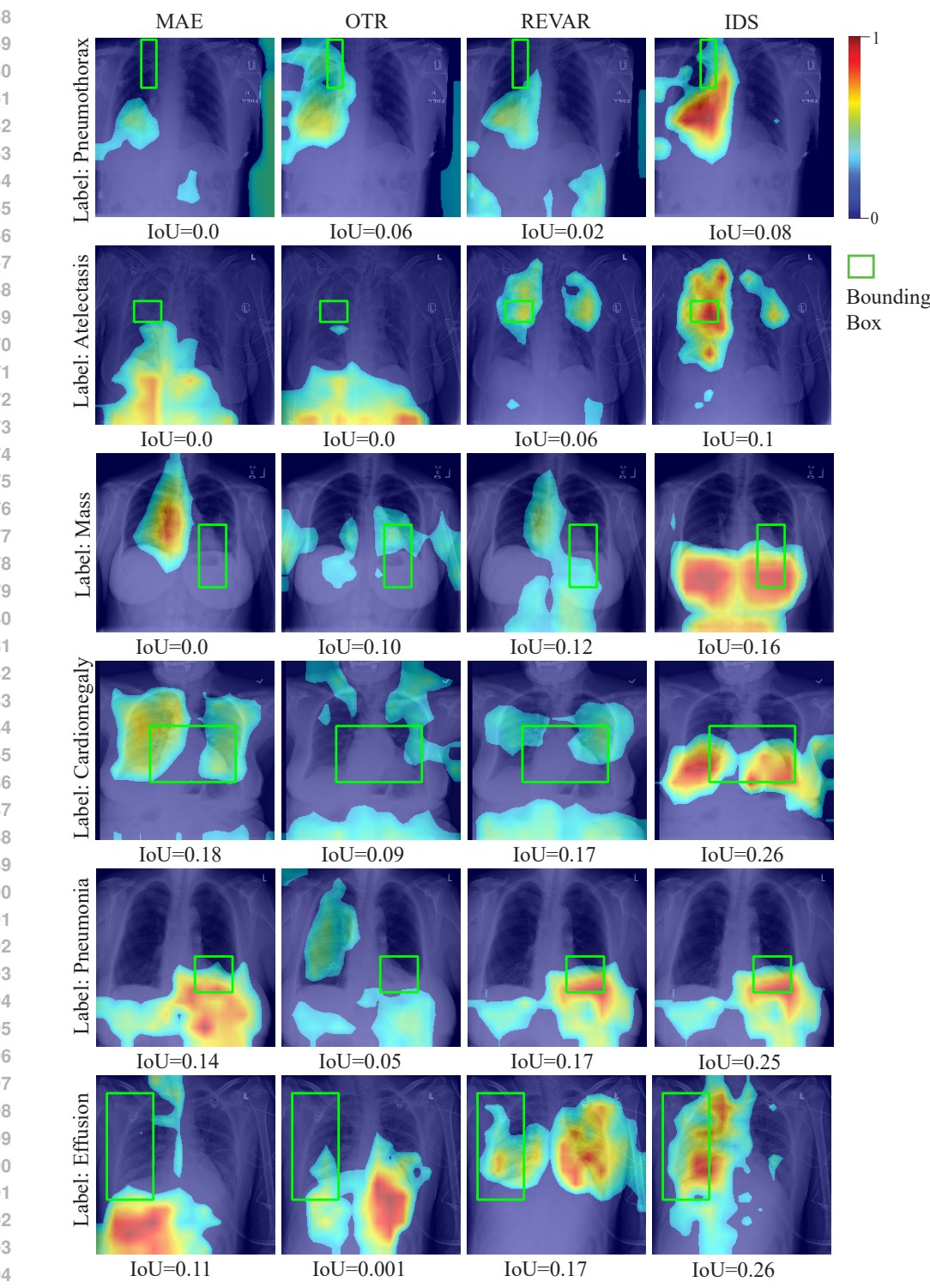

Figure 5: Grad-CAM visualization results on NIH-ChestX-ray14 dataset for various disease labels including Pneumothorax, Atelectasis, Mass, Cardiomegaly, Pneumonia, and Effusion. The visualizations from MAE (Xiao et al., 2023), OTR (Guo et al., 2022), REVAR (Jain et al., 2024), and IDS are shown in the first, second, third, and fourth columns, respectively. The green bounding boxes represent the ground truth regions of interest for each label, and the corresponding IoU score is shown below each image, which quantifies the overlap between the Grad-CAM heatmap and the ground truth bounding box. For each Grad-CAM visualization, higher IoU scores indicate a better localization of the activated regions in relation to the ground truth.

