# OpenReview forum: "Informative Data Selection for Thorax Disease Classification"
_ICLR.cc/2025/Conference — Submitted to ICLR 2025_

### Official Review · Reviewer_Y6VF · 2024-10-26

**Soundness:** 2
**Presentation:** 3
**Contribution:** 2
**Rating:** 5
**Confidence:** 4

**Summary:**

This paper proposes an Informative Data Selection (IDS) method to re-weight synthetic images generated by generative models based on an information-theoretic measure, the Information Bottleneck (IB). IDS trains a sample re-weighting network to minimize the IB loss on the synthetic data. This approach better adheres to the IB principle by encouraging the learning of features that are more correlated with the outputs and less correlated with the inputs. The authors conducted experiments on three thorax disease classification datasets (CheXpert, COVIDx, and NIH ChestX-ray14). They compared their method against other data selection and sample re-weighting techniques, showing superior results.

**Strengths:**

+ The use of the IB principle for sample re-weighting sounds interesting.
+ The paper shows promising results based on the evaluation using three significant thorax disease classification benchmarks compared to state-of-the-art methods.
+ The paper is well-organized and easy to follow.

**Weaknesses:**

1- Regarding methodology, my main comments are lack of sufficient details:

- The computation of gradients for parameters of the classifier and sample re-weighting in Steps 4 and 5 is not detailed. For example, how are the gradients of the VIB loss with respect to θ computed? Are there any approximations to handle the expectations in the IB loss?
- Step 3 refers to computing the class centroids using Equation (3), which involves both real and synthetic data. However, the notation is dense, and the practical implementation of this step, especially with large datasets, could be challenging. Can you elaborate on details of how centroids are updated during training?
- Also, in practice, Bi-level optimization can be sensitive to hyperparameters. Maybe providing discussion (e.g., choice of learning rates, use of momentum) for a better understanding

2- Regarding computational complexity:

- Maybe I missed it, but I could not find a discussion on the computational overhead introduced by the sample re-weighting network and the IB loss minimization, especially when scaling to larger datasets or higher-resolution images.
- Generating synthetic data using diffusion models can be time-consuming. The authors should elaborate on the efficiency of the data generation process and its impact on the overall training time.

3- About Practical Constraints

- The method assumes that the diffusion models can generate sufficiently realistic and diverse synthetic images. However, the limitations of the synthetic data, such as potential biases introduced during generation, have been limitedly analyzed.
- Also, since IDS relies on pre-trained VAEs and diffusion models, its performance may be sensitive to the quality of these models; I think analysis is required to see their impact.

4- Regarding experiments

- The paper would benefit from more extensive ablation studies to isolate the effects of different components of IDS. For example, analyzing the impact of the VIB term separately from the re-weighting network. In addition, the results are presented without discussion of statistical significance.

Last minor comment about clinical relevance: I suggest authors include a discussion of how IDS might impact the clinical routine of decision-making or how these weights correlate with specific data characteristics.

**Questions:**

- While the authors reference the standard bi-level optimization approach, I suggest including a brief explanation or highlighting any adaptations made that would improve clarity.
- Additional ablation studies are suggested (see above). Discuss the statistical significance to substantiate performance claims.
- Address the computational costs associated with IDS, including training and inference times

---

> ### Author Response · Authors · 2024-12-02
> **Response to Reviewer Y6VF Part 1**
>
> We appreciate the review and the suggestions in this review. The raised issues are addressed below.
>
> **Responses to the Weaknesses**
>
> **1. (1) “The computation of gradients for parameters of the classifier and sample re-weighting in Steps 4 and 5 is not detailed...”**
>
> The variational upper bound of the IB loss, VIB, is defined in Equation (2) in Section 3.2 of our paper.  The VIB loss is differentiable w.r.t.  the class centroids of the input features and the learned features. Furthermore, the class centroids of the input features and the learned features defined in Equation (3) in Section 3.3 of our paper are differentiable w.r.t., the class centroids of the parameters of the sample re-weighting network $\theta$ and the parameters of the classifier $\Theta$. Therefore, the gradients of the VIB loss w.r.t. $\theta$ and $\Theta$ can be computed by the back-propagation algorithm based on the chain rule.
>
> **1. (2) “... Can you elaborate on details of how centroids are updated during training?”**
>
> The centroids of the input features and the learned features are computed based on Equation (3) in Section 3.3 of our paper. The centroids are updated at the beginning of each epoch as described in Algorithm 3 in Section C of the appendix in our paper.
>
> **1. (3) “... Bi-level optimization can be sensitive to hyperparameters. Maybe providing discussion (e.g., choice of learning rates, use of momentum)...”**
>
> We have added discussions about the hyperparameters to Section D.1 of the revised paper. The initial learning rate is determined through 5-fold cross-validation on the training set for each model and dataset. The weight decay is set to 0.05, and the momentum parameters $\beta_1$ and $\beta_2$ are set to 0.9 and 0.999 for all the experiments.
>
> **2. (1) “discussion on the computational overhead introduced by the sample re-weighting network and the IB loss minimization...”**
>
> To evaluate the efficiency of different components in the IDS, we compare the disease classification performance and the training time of the baseline model ViT-B, the IDS model IDS-ViT-B, and two ablation models, which are IDS-ViT-B without VIB and IDS-ViT-B without the re-weighting network. The comparison is performed on the COVIDx dataset. The training time is evaluated on four NVIDIA A100 GPUs. The results are shown in the table below. With only a $30\%$ increase in the training time, IDS-ViT-B improves the classification accuracy on COVIDx by $2.0\%$, demonstrating the effectiveness of integrating these components of IDS into the baseline model. The study further confirms the individual contributions of the VIB and the re-weighting network, underlining the importance of both components in enhancing model performance while maintaining a manageable increase in computational demand. The training efficiency analysis has been added to Section D.4 in the appendix of the revised paper.
>
> | Methods           | COVIDx (Accuracy) | Training Time (minutes/epoch) |
> |---|---|---|
> | ViT-B           | 95.3      | 2.6         |
> | IDS-ViT-B w/o VIB       | 96.4      | 3.2         |
> | IDS-ViT-B w/o Re-weighting Network  | 96.7      | 2.8         |
> | IDS-ViT-B         | **97.3**    | 3.4         |
>
>
> **2. (2) “... the efficiency of the data generation process...”**
>
> To evaluate the efficiency of the diffusion model used for the data generation in the IDS, we compare the performance data generation time of IDS-ViT-B using three different diffusion models for data generation, which are DiT-B, DiT-L, and DiT-XL [4].
> The data generation time and the classification accuracy of the IDS model with different diffusion models on the COVIDx dataset are shown in the table below. It is observed that the synthetic data generation process with the diffusion models in IDS is efficient, with around $0.01$ seconds/image using the fastest diffusion model. In addition, recent works on accelerating the diffusion models [1, 2, 3], which are orthogonal to our work, can be employed for our IDS to accelerate the data generation process.
>
> | Methods     | COVIDx (Accuracy) | Generation Time (seconds/image) |
> |---|---|---|
> | ViT-B     | 95.3      | -           |
> | IDS-ViT-B (DiT-B) | 97.1    | 0.095         |
> | IDS-ViT-B (DiT-L) | **97.3**    | 0.151         |
> | IDS-ViT-B (DiT-XL)| **97.3**    | 0.176         |

---

> > ### Author Response · Authors · 2024-12-02
> > **Response to Reviewer Y6VF Part 2**
> >
> > **3. (1) “...the limitations of the synthetic data, such as potential biases introduced during generation, have been limitedly analyzed....”**
> >
> > We agree with the reviewer that there are limitations to the use of synthetic data, which also serves as the motivation for performing data selection in generative data augmentation (GDA) with the proposed IDS, as discussed in Section 1 and Section 2.3 of our paper.
> > The synthetic data generated by the diffusion models may contain bias and noise, leading to discrepancies between the labels and the features of the synthetic data [4]. In the GDA process, the conditional latent diffusion models generate the features of synthetic data using labels as the conditional inputs. Due to inherent randomness in the generation process of the diffusion models, previous studies [5, 6, 7, 8] have observed that conditional diffusion models may produce features that are misaligned with the given labels, which results in performance degradation in classification models trained on such synthetic data. To address this issue, we propose a principled sample re-weighting method, which performs data selection in the synthetic data generated by the diffusion models for the GDA.
> >
> > **3. (2) “... IDS relies on pre-trained VAEs and diffusion models, its performance may be sensitive to the quality of these models...”**
> >
> > In our response to Weakness 2. (2), we have compared the performance of the IDS models using three different diffusion models for data generation, which are DiT-B, DiT-L, and DiT-XL [4]. It is observed that the performance of the IDS model is not sensitive to the selection of the diffusion models used for data generation. The IDS-ViT-B based on the largest DiT model DiT-XL only outperforms the IDS-ViT-B based on the smallest DiT model DiT-B by 0.2% in classification accuracy on COVIDx, demonstrating the merit of IDS in mitigating the noise in the synthetic data generated by less powerful diffusion models.
> >
> > **4. (1) “... more extensive ablation studies to isolate the effects of different components of IDS...”**
> >
> > In our response to Weakness 2. (1), we have compared the disease classification performance of the baseline model ViT-B, the IDS model IDS-ViT-B, and two ablation models, which are IDS-ViT-B without VIB and IDS-ViT-B without the re-weighting network.  The comparison is performed on the COVIDx dataset. The results demonstrate that both the VIB and the re-weighting network contribute to the improved performance of the IDS model over the base model.
> >
> > **Responses to the Questions**
> >
> > **1. “...standard bi-level optimization approach, I suggest including a brief explanation ...”**
> >
> > We have added an intuitive explanation of the bi-level optimization framework at the beginning of the paragraph describing the optimization of the IDS in Section 3.3 of the revised paper.
> > The bi-level optimization process alternates between updating the parameters of the sample re-weighting network and the parameters of the classifier, leveraging the gradient-based methods to efficiently manage the interdependencies between the two tasks.
> > In the bi-level optimization process, the lower level optimizes a sample re-weighting network to assign importance weights to training samples, enhancing classifier training. The upper level then trains the classifier with the re-weighted samples for improved classification performance.

---

> ### Author Response · Authors · 2024-12-02
> **Response to Reviewer Y6VF Part 3**
>
> **2. “Additional ablation studies are suggested (see above). Discuss the statistical significance to substantiate performance claims.”**
>
> Please refer to our response to Weakness 2. (1) for the ablation study results.
>
> To assess whether the improvements by our proposed IDS are statistically significant and out of the range of error, we train both the IDS models and the leading baseline methods on different datasets from Table 1, Table 2, and Table 3 for $10$ times with different seeds for random initialization of the networks and train/val/test splits. Subsequently, we perform the t-test between the results of the IDS models and the results of the best baseline methods on different datasets.
> The mean and standard deviation of the results and the p-values of the t-test are shown in the table below, which is also added to Section D.3 of the appendix of the revised paper. The t-test results suggest that the improvements of IDS over the baseline methods are statistically significant with $p\ll 0.05$, and it is not caused by random error.
>
> | Dataset   | Architecture | CheXpert (mAUC)     | COVIDx (Accuracy)   | NIH ChestX-ray14 (mAUC) |
> |---|---|---|---|---|
> | Best Baseline | ViT-S/16   | $89.2 \pm 0.067$    | $96.2 \pm 0.122$    | $82.3 \pm 0.045$    |
> | IDS     |   -   | $89.6 \pm 0.112$    | $97.1 \pm 0.125$    | $82.7 \pm 0.052$    |
> | p-value   | -    | $1.44 \times 10^{-10}$  | $3.20 \times 10^{-12}$  | $4.07 \times 10^{-13}$  |
> | Best Baseline | ViT-B/16   | $89.3 \pm 0.045$    | $96.3 \pm 0.158$    | $83.0 \pm 0.051$    |
> | IDS     |  -    | $90.1 \pm 0.096$    | $97.3 \pm 0.136$    | $83.4 \pm 0.065$    |
> | p-value   | -    | $1.24 \times 10^{-15}$  | $1.40 \times 10^{-11}$  | $1.48 \times 10^{-12}$  |
>
>
> **3. “Address the computational costs associated with IDS…”**
>
> Please refer to our response to Weakness 2. (1) and Weakness 2. (2)  for the analysis of the computational costs in both the training process and the data generation process.
>
>
> **References**
>
> [1] Li, Lijiang, et al. "Autodiffusion: Training-free optimization of time steps and architectures for automated diffusion model acceleration." CVPR 2023.
>
> [2] Ma, Xinyin, Gongfan Fang, and Xinchao Wang. "Deepcache: Accelerating diffusion models for free." CVPR 2024.
>
> [3] Shang, Yuzhang, et al. "Post-training quantization on diffusion models." CVPR 2023.
>
> [4] Peebles, William, and Saining Xie. "Scalable diffusion models with transformers." CVPR 2023.
>
> [5] Azizi, Shekoofeh, et al. "Synthetic data from diffusion models improves imagenet classification." TMLR 2023.
>
> [6] Trabucco, Brandon, et al. "Effective data augmentation with diffusion models." ICLR 2024.
>
> [7] Sarıyıldız, Mert Bülent, et al. "Fake it till you make it: Learning transferable representations from synthetic imagenet clones." CVPR 2023.
>
> [8] He, Ruifei, et al. "Is synthetic data from generative models ready for image recognition?." ICLR 2023

---

### Official Review · Reviewer_iQBZ · 2024-11-02

**Soundness:** 2
**Presentation:** 2
**Contribution:** 2
**Rating:** 6
**Confidence:** 4

**Summary:**

This paper introduces a novel method called Informative Data Selection (IDS) to improve thorax disease classification by addressing challenges inherent in Generative Data Augmentation (GDA). While GDA leverages generative models like diffusion models to create synthetic training data—particularly valuable in medical fields with limited high-quality annotated data—it often introduces noise that can degrade model performance. IDS mitigates this issue by applying the Information Bottleneck (IB) principle to re-weight synthetic samples, emphasizing informative data to enhance the classifier’s accuracy.

**Strengths:**

The paper presents an innovative method, IDS, which effectively addresses the scarcity of annotated training data for lung images by assigning higher weights to more informative synthetic images. This approach shows promising potential for practical clinical applications.
The authors have provided the code, offering valuable resources for the community to learn from and build upon, which enhances reproducibility and fosters collaborative advancement in the field.
Comprehensive Related Work: The related work section is thorough, detailing the challenges of obtaining high-quality annotations in the medical field and providing an in-depth analysis of the noise introduced by synthetic data from generative models.
The paper includes extensive experiments on three datasets—CheXpert, COVIDx, and NIH-ChestXray-14—that validate the effectiveness of IDS, supported by a robust amount of experimental data.

**Weaknesses:**

The metric mAUC used in the experiments lacks a detailed formula or explanation, which may hinder readers’ understanding of the evaluation process.
Section 3.2 contains numerous embedded formulas within the main text, making it less intuitive to read. Reorganizing this section by presenting formulas separately and providing explanatory text could enhance readability.
Placement of Algorithm 1: While Algorithm 1 is clearly described, placing it in the appendix could streamline the main text and allow readers to focus on the conceptual framework before diving into algorithmic details.
Discussion on Noise Handling: The method effectively handles noisy and imbalanced datasets; however, the paper lacks a specific discussion on the generation and nature of the noise, as well as how IDS mitigates its effects.
Comparison of Computational Efficiency: The authors focus on accuracy comparisons but do not address efficiency and computational costs. Including this analysis would provide a more comprehensive understanding of IDS’s practicality in real-world applications.

**Questions:**

1.	Why were ViT-S, ViT-B, and DN121 selected as the base classification networks? Is there a particular reason for not evaluating the performance on larger models like ViT-L?
	2.	While the paper focuses on lung classification tasks, could the IDS method be extended to other lung-related tasks such as segmentation or lesion detection? Exploring this could demonstrate the versatility of the approach.
	3.	The experiments utilize classification baseline models based on Transformer and CNN architectures. Have you considered including recent State Space Models, such as Vision Mamba, in your experiments to assess IDS’s effectiveness across different model architectures?
	4.	In practical applications, how would you recommend users balance data selection efficiency with computational costs? Providing guidelines or strategies based on your findings could be beneficial for practitioners.

---

> ### Author Response · Authors · 2024-12-02
> **Response to Reviewer iQBZ Part 1**
>
> We appreciate the review and the suggestions in this review. The raised issues are addressed below.
>
> **Responses to the Weaknesses**
>
> **1. “The metric mAUC used in the experiments lacks a detailed formula or explanation...”**
>
> We have added details on the mean Area Under the Curve (mAUC) metric in Section 4.1 of the revised paper. The mean Area Under the Curve (mAUC) is adopted as the evaluation metric for the multi-label disease classification datasets CheXpert and NIH ChestX-ray14 following [6]. In multi-label disease classification, the prediction on each label can be considered as a binary classification task. We first compute the individual Area Under the Curve (AUC) for each label by calculating the area under the receiver operating characteristic curve [11], which provides a measure of the performance of the model in detecting each disease. The mean Area Under the Curve (mAUC) is then computed by averaging the individual Area Under the Curve (AUC) values calculated for each disease label. This approach aggregates the performance across all labels, thereby furnishing a comprehensive metric that reflects the overall performance of the model.
>
> **2. “Section 3.2 contains numerous embedded formulas within the main text, making it less intuitive to read...”**
>
> Below is an intuitive explanation of Section 3.2. To optimize the IB loss for the data re-weighting in IDS, we first derive a variational upper bound for the IB loss, which can be minimized by standard SGD algorithms. The derivation begins by formulating the IB loss with the input features, learned features, and target labels. To achieve that, we first define the calculation of the class centroids for both the input and learned features, which are used to assign the class membership probabilities to each sample. These probabilities are then utilized to calculate the mutual information between the input feature and the learned feature and the mutual information between the learned feature and the target label, which are then used to compute the IB loss. Theorem 3.1 provides a variational upper bound for the IB loss, facilitating its optimization with standard SGD algorithms.
>
> **3. “Placement of Algorithm 1... ”**
>
> Algorithm 1 has been moved to Section C of the appendix in the revised paper.
>
> **4. “ Discussion on Noise Handling...”**
>
> The noise in the synthetic data generated by diffusion models primarily arises from discrepancies between the labels and the features of the synthetic data [4]. In generative data augmentation, latent diffusion models produce the features of synthetic data using labels as conditional inputs. Due to the inherent randomness in the generation process of the diffusion models, existing works [1,2,3,5] have observed that conditional diffusion models may produce features misaligned with the given labels, which results in performance degradation in classification models trained on such synthetic data.

---

> > ### Author Response · Authors · 2024-12-02
> > **Response to Reviewer iQBZ Part 2**
> >
> > **5. “Comparison of Computational Efficiency...”**
> >
> > We have added computational efficiency analysis on the training process and the data generation process of IDS in Section D.4 and Section D.5. The efficiency analysis is also attached below.
> >
> > To evaluate the efficiency of the IDS, we assess the training time of the baseline model ViT-B, the IDS model IDS-ViT-B, and two ablation models, which are IDS-ViT-B without VIB and IDS-ViT-B without the re-weighting network. The comparison is performed on the COVIDx dataset. The training time is assessed on four NVIDIA A100 GPUs. The results are shown in the table below. With only a $30\%$ increase in the training time, IDS-ViT-B improves the classification accuracy of the base model ViT-B by $2.0\%$ on COVIDx, demonstrating the effectiveness of integrating these components into the baseline model. The study further confirms the individual contributions of the VIB and the re-weighting network, underlining the importance of both components in enhancing model performance while maintaining a manageable increase in computational demand.
> >
> > | Methods           | COVIDx (Accuracy) | Training Time (minutes/epoch) |
> > |---|---|---|
> > | ViT-B           | 95.3      | 2.6         |
> > | IDS-ViT-B w/o VIB       | 96.4      | 3.2         |
> > | IDS-ViT-B w/o Re-weighting Network  | 96.7      | 2.8         |
> > | IDS-ViT-B         | **97.3**    | 3.4         |
> >
> > To evaluate the efficiency of the diffusion model used for the data generation in the IDS, we compare the performance and the data generation time of the IDS-ViT-B using three different diffusion models for data generation, which are DiT-B, DiT-L, and DiT-XL [7].
> > The data generation time and the classification accuracy of the IDS model with different diffusion models on the COVIDx dataset are shown in the Table below. It is observed that the synthetic data generation process with the diffusion models in IDS is efficient, with around $0.01$ seconds/image using the fastest diffusion model. In addition, recent works on accelerating the diffusion models [8, 9, 10], which are orthogonal to our work, can be employed for our IDS for faster data generation.
> >
> > | Methods     | COVIDx (Accuracy) | Generation Time (seconds/image) |
> > |---|---|---|
> > | ViT-B     | 95.3      | -           |
> > | IDS-ViT-B (DiT-B) | 97.1    | 0.095         |
> > | IDS-ViT-B (DiT-L) | **97.3**    | 0.151         |
> > | IDS-ViT-B (DiT-XL)| **97.3**    | 0.176         |
> >
> >
> >
> >
> > **Responses to the Questions**
> >
> > **1. “Why were ViT-S, ViT-B, and DN121 selected as the base classification networks?”**
> >
> > We select ViT-S, ViT-B, and DN121 as the base classification networks following the state-of-the-art method for thorax disease classification, MAE [6], which is the baseline of our method.
> >
> > **2. “...could the IDS method be extended to other lung-related tasks such as segmentation or lesion detection...”**
> >
> > IDS can be extended to classification-oriented lung-related tasks such as lesion detection. Since lesion detection is indeed a binary classification task, the proposed IDS can be used to generate synthetic data in both the positive and the negative classes for lesion detection and re-weigh the generated synthetic data during the training process.
> >
> >
> > **3. “The experiments utilize classification baseline models based on Transformer and CNN architectures...”**
> >
> > We select CNNs and Transformers as the baseline models in our work following the state-of-the-art method for thorax disease classification, MAE [6]. We will further investigate the effectiveness of IDS on recent State Space Models, such as Vision Mamba, in the final version of our paper, as suggested by the reviewer.
> >
> >
> > **4. “In practical applications, how would you recommend users balance data selection efficiency with computational costs? ...”**
> >
> > To balance the efficiency of the IDS and the computational costs in the practical application, the user can reduce the number of synthetic data generated for IDS. As described in Section 3.1 of our paper, we set the synthetic labels to be the same as the labels of the original training set. Let $N$ be the size of the original training set, then the number of synthetic data is also $N$. In the practical scenario, the user can reduce the number of synthetic data to smaller values, such as $N/8$, $N/4$, and $N/2$, by generating the synthetic data with a subset of the labels for the training set. As shown in the ablation study in the table below, IDS-ViT-B still outperforms ViT-B by $1.2\%$ in classification accuracy on COVIDx, even with $N/8$ synthetic data.
> >
> > | Methods     | Synthetic Data Size | COVIDx (Accuracy) |
> > |---|---|---|
> > | ViT-B   |  0  | 95.3     |
> > | IDS-ViT-B | $N/8$ | 96.5          |
> > | IDS-ViT-B | $N/4$ | 96.9         |
> > | IDS-ViT-B |  $N/2$ | 97.2          |
> > | IDS-ViT-B |  $N$ | 97.3          |
> >
> > In addition, users can reduce the efficiency of IDS by applying acceleration methods on the diffusion models [8, 9, 10], which are orthogonal to our work.

---

> > > ### Author Response · Authors · 2024-12-02
> > > **Response to Reviewer iQBZ Part 3**
> > >
> > > **References**
> > >
> > > [1] Azizi, Shekoofeh, et al. "Synthetic data from diffusion models improves imagenet classification." TMLR 2023.
> > >
> > > [2] Trabucco, Brandon, et al. "Effective data augmentation with diffusion models." ICLR 2024.
> > >
> > > [3] Sarıyıldız, Mert Bülent, et al. "Fake it till you make it: Learning transferable representations from synthetic imagenet clones." CVPR 2023.
> > >
> > > [4] Na, Byeonghu, et al. "Label-Noise Robust Diffusion Models." ICLR 2024.
> > >
> > > [5] He, Ruifei, et al. "Is synthetic data from generative models ready for image recognition?." ICLR 2023
> > >
> > > [6] Xiao, Junfei, et al. "Delving into masked autoencoders for multi-label thorax disease classification." WACV 2023.
> > >
> > > [7] Peebles, William, and Saining Xie. "Scalable diffusion models with transformers." CVPR 2023.
> > >
> > > [8] Li, Lijiang, et al. "Autodiffusion: Training-free optimization of time steps and architectures for automated diffusion model acceleration." CVPR 2023.
> > >
> > > [9] Ma, Xinyin, Gongfan Fang, and Xinchao Wang. "Deepcache: Accelerating diffusion models for free." CVPR 2024.
> > >
> > > [10] Shang, Yuzhang, et al. "Post-training quantization on diffusion models." CVPR 2023.
> > >
> > > [11] Bradley, Andrew P. "The use of the area under the ROC curve in the evaluation of machine learning algorithms." Pattern recognition 1997

---

### Official Review · Reviewer_Q4Pt · 2024-11-03

**Soundness:** 2
**Presentation:** 3
**Contribution:** 2
**Rating:** 3
**Confidence:** 4

**Summary:**

The authors proposed a data augmentation method by assigning weights to synthetic data. The synthetic data are generated based on a diffusion model, while the data re-weighting is computed based on the information bottleneck. Public chest X-ray datasets are adopted for the experiments and evaluation, where prior data selection and reweighting methods are included in the comparison study. Only marginal performance improvements of the proposed method are reported. The manuscript is overall easy to follow. However, it also suffers from several flaws, which are detailed below.

**Strengths:**

+ Important data selection and reweighting strategy for model training is investigated.
+ The manuscript is easy to follow overall.

**Weaknesses:**

- The comparison study is conducted between the proposed method and previous data selection and sample reweighting methods. However, the proposed method is indeed a data augmentation method with add-on synthetic data as extra training data. Previous and common data augmentation methods (especially those using synthetic data) should be included in the comparison.

- The current sample reweighting is only applied to the synthetic data. How about applying the reweighting strategy to the real data as well? How will that affect the training speed and performance?

- The reported performance gain of the proposed method is really marginal (often less than 1-2% in classification metrics). Such performance gain further raise the questions about the performance gap of the proposed method to common data augmentation methods.

- The contribution section could be better summarized.

- The presented results for localization on cardiomegaly are not necessary and are not good example showing relation between weights and localization results. Cardiomegaly is about the heart, and all the bounding boxes will be around the heart. I did not how that could reveal the benefit of reweighting.

- The proposed method is also strongly related to the active learning methods (using selective synthetic data instead of real ones). The authors could consider prior arts in this field for the related work and comparison study.

**Questions:**

See above.

---

> ### Author Response · Authors · 2024-12-02
> **Response to Reviewer Q4Pt Part 1**
>
> We appreciate the review and the suggestions in this review. The raised issues are addressed below.
>
> **Responses to the Weaknesses**
>
> **1. “...Previous and common data augmentation methods (especially those using synthetic data) should be included ...”**
>
> In our experiments in Section 4.2, we have compared IDS with a baseline method CBF [1], which performs Generative Data Augmentation (GDA) by incorporating synthetic data generated by diffusion models into the training set. To mitigate the potential noise in synthetic GDA, CBF proposes a CLIP Filter (CF) strategy to rule out low-quality synthetic data. In addition, we also included a baseline that performs vanilla GDA [4] on the base model without data selection.
> As evidenced in Table 1, Table 2, and Table 3 of the revised paper, our proposed IDS significantly outperforms the CBF and the vanilla GDA on all the datasets.
>
> **2. “...How about applying the reweighting strategy to the real data as well? ...”**
>
> We appreciate the suggestions from the reviewer and have added experiments applying IDS to re-weight both the real data and the synthetic data. The results are shown in the table below and have been added to Table 1, Table 2, and Table 3 of the revised paper. It is observed that applying IDS to re-weighting both the real data and the synthetic data further boosts the performance of the IDS models. For example, the IDS-ViT-B re-weighting both the synthetic data and the real data outperforms the IDS-ViT-B re-weighting only the synthetic data by 0.6% in mAUC on CheXpert, demonstrating the merits of IDS in selecting informative samples in both the real dataset and the synthetic dataset.
>
> | Dataset    | Method           | mAUC / Accuracy |
> | --- | ---- | --- |
> | CheXpert   | ViT-B          | 89.3    |
> |      | IDS-ViT-B          | 90.1    |
> |      | IDS-ViT-B (Re-weighting Real Data) | **90.7**    |
> | COVIDx     | ViT-B          | 95.3    |
> |      | IDS-ViT-B          | 97.3    |
> |      | IDS-ViT-B (Re-weighting Real Data) | **97.7**    |
> | NIH ChestX-ray14 | ViT-B          | 83.0    |
> |      | IDS-ViT-B          | 83.4    |
> |      | IDS-ViT-B (Re-weighting Real Data) | **83.9**    |
>
> **3. “...performance gain of the proposed method is really marginal ...”**
>
> To demonstrate that the improvements by our proposed IDS are statistically significant and not marginal, we have trained both the IDS models and the leading baseline methods on different datasets from Table 1, Table 2, and Table 3 for $10$ times with different seeds for random initialization of the networks and train/val/test splits. Subsequently, we have performed the t-test between the results of the IDS models and the results of the best baseline methods on different datasets.
> The mean and standard deviation of the results and the p-values of the t-test are shown in the table below, which is also added to Section D.3 of the appendix of the revised paper. The t-test results suggest that the improvements of IDS over the baseline methods are statistically significant with $p\ll 0.05$, and it is not caused by random error.
>
> | Dataset   | Architecture | CheXpert (mAUC)     | COVIDx (Accuracy)   | NIH ChestX-ray14 (mAUC) |
> |---|---|---|---|---|
> | Best Baseline | ViT-S/16   | $89.2 \pm 0.067$    | $96.2 \pm 0.122$    | $82.3 \pm 0.045$    |
> | IDS     |   -   | $89.6 \pm 0.112$    | $97.1 \pm 0.125$    | $82.7 \pm 0.052$    |
> | p-value   | -    | $1.44 \times 10^{-10}$  | $3.20 \times 10^{-12}$  | $4.07 \times 10^{-13}$  |
> | Best Baseline | ViT-B/16   | $89.3 \pm 0.045$    | $96.3 \pm 0.158$    | $83.0 \pm 0.051$    |
> | IDS     |  -    | $90.1 \pm 0.096$    | $97.3 \pm 0.136$    | $83.4 \pm 0.065$    |
> | p-value   | -    | $1.24 \times 10^{-15}$  | $1.40 \times 10^{-11}$  | $1.48 \times 10^{-12}$  |
>
>
> In addition, re-weighting both the real and the synthetic data further boosts the performance of IDS, as demonstrated in the response to Weakness 2.
>
> **4. “The contribution section could be better summarized.”**
>
> We have improved the summarization of the contributions of our paper on Lines 117-120 in Section 1 of the revised paper.

---

> > ### Author Response · Authors · 2024-12-02
> > **Response to Reviewer Q4Pt Part 2**
> >
> > **5. “The presented results for localization on cardiomegaly are not necessary...”**
> >
> > We respectfully disagree with the point that the "results for localization on cardiomegaly are not necessary". Although all the bounding boxes are around the heart, the alignment between the bounding box and the disease localization area predicted by applying GradCAM on a pre-trained DNN is a good indicator of the quality of the synthetic data. A synthetic image with a higher IoU between the bounding box and the disease localization area is considered a more informative sample for this disease as a larger portion of the predicted disease localization area overlaps with the ground-truth bounding box of the disease. As shown in the figures in the first row of Figure 1 in our paper, there are synthetic images whose localization area is outside the ground-truth bounding box, demonstrating the necessity of data selection in the synthetic data. In addition, the figures in the second row of Figure 1 in our paper show that there is a stronger positive correlation between the IoU scores and the importance weights by our IDS than that for competing baselines, demonstrating the superiority of IDS for assigning higher importance weights to more informative synthetic images.
> >
> >
> >
> > **6. “The proposed method is also strongly related to the active learning methods...”**
> >
> > We have included the discussions on the related works on active learning and performed a comparison between the IDS and the active learning methods for the data selection on the synthetic data in Section D.6 in the appendix of the revised paper.
> >
> > Active learning (AL) methods aim to minimize the effort required for labeling training data by strategically choosing the most informative instances for annotation, which can be adapted to identify the informative synthetic data. To show the advantages of the IDS over the active learning methods in selecting the most informative synthetic data, we compare the IDS with two state-of-the-art active learning methods, which are CAMPAL [2] and SAAL [3]. Both CAMPAL and SAAL are adopted to select samples from the synthetic dataset generated by the diffusion models to add to the training set. The results are shown in the table below. It is observed that the IDS outperforms the competing active learning methods on all the datasets, demonstrating the superiority of the IDS in selecting informative training samples compared to the active learning methods.
> >
> > | Methods  | COVIDx (mAUC) | Covid-19 (Accuracy) | NIH ChestX-ray14 (mAUC) |
> > | --- | --- | --- | --- |
> > | ViT-B    | 89.3    | 95.3      | 83.0        |
> > | CAMPAL-ViT-B | 89.4    | 96.2      | 83.0        |
> > | SAAL-ViT-B | 89.3    | 95.9      | 83.1      |
> > | IDS-ViT-B  | **89.6**  | **97.3**    | **83.4**      |
> >
> > **References**
> >
> > [1] He, Ruifei, et al. "Is synthetic data from generative models ready for image recognition?." ICLR 2023
> >
> > [2] Yang, Jianan, et al. "Towards controlled data augmentations for active learning." International Conference on Machine Learning. PMLR, 2023.
> >
> > [3] Kim, Yoon-Yeong, et al. "Saal: sharpness-aware active learning." International Conference on Machine Learning. PMLR, 2023.
> >
> > [4] Azizi, Shekoofeh, et al. "Synthetic data from diffusion models improves imagenet classification." TMLR 2023.

---

> > > ### Comment · Reviewer_Q4Pt · 2024-12-03
> > >
> > > I thank the authors for the detailed response to my comments, which addressed some of my concerns. However, I will keep my rate by considering the overall novelty and performance gain of the proposed method.

---

### Official Review · Reviewer_ntDN · 2024-11-04

**Soundness:** 1
**Presentation:** 1
**Contribution:** 1
**Rating:** 3
**Confidence:** 4

**Summary:**

This work presents a weighting scheme for samples while training a model with synthetic data to take into account more informative samples.

They train an external network called the "sample re-weighting network". I believe this external network is trained (to state information bottleneck it as bluntly as possible) such that it minimizes the mutual information between embeddings of synthetic images and the respective images and then maximizes mutual information between embeddings of synthetic images and those image labels. This is performed to improve the diversity of generated samples.

I'm not very clear if the synthetic samples are generated dynamically during training or if the samples are already generated and have a weight applied. Maybe having a figure would make this more clear.

**Strengths:**

.

**Weaknesses:**

The comparisons are not conclusive. Specifically, there is no significance testing or estimate of variance so nothing can be concluded from these results. The datasets are not so big that this is intractable. This is specifically important to have for this work because the numbers are so close.

I don't share the intuition that this method works, so it should be asserted by the authors using a significance test that I am wrong. I do not believe sufficient evidence is presented to support the claims of significance in this work.

You can achieve this by randomizing train/val/test splits and reporting the mean test performance and the stdev.

**Questions:**

Why are the experiments limited to chest X-ray data? Has this already been done on other image tasks? Perhaps something common like faces or animals?

---

> ### Author Response · Authors · 2024-12-02
> **Response to Reviewer ntDN**
>
> We appreciate the review and the suggestions in this review. The raised issues are addressed below.
>
> **Responses to the Weaknesses**
>
> **1. “...there is no significance testing or estimate of variance...”**
>
> To assess whether the improvements by our proposed IDS are statistically significant and out of the range of error, we have trained both the IDS models and the leading baseline methods on different datasets from Table 1, Table 2, and Table 3 for $10$ times with different seeds for random initialization of the networks and train/val/test splits. Subsequently, we have performed the t-test between the results of the IDS models and the results of the best baseline methods on different datasets.
> The mean and standard deviation of the results and the p-values of the t-test are shown in the table below, which has also been added to Section D.3 of the appendix in the revised paper. The t-test results suggest that the improvements of IDS over the baseline methods are statistically significant with $p\ll 0.05$, and it is not caused by random error.
>
> | Dataset   | Architecture | CheXpert (mAUC)     | COVIDx (Accuracy)   | NIH ChestX-ray14 (mAUC) |
> |---|---|---|---|---|
> | Best Baseline | ViT-S/16   | $89.2 \pm 0.067$    | $96.2 \pm 0.122$    | $82.3 \pm 0.045$    |
> | IDS     |   -   | $89.6 \pm 0.112$    | $97.1 \pm 0.125$    | $82.7 \pm 0.052$    |
> | p-value   | -    | $1.44 \times 10^{-10}$  | $3.20 \times 10^{-12}$  | $4.07 \times 10^{-13}$  |
> | Best Baseline | ViT-B/16   | $89.3 \pm 0.045$    | $96.3 \pm 0.158$    | $83.0 \pm 0.051$    |
> | IDS     |  -    | $90.1 \pm 0.096$    | $97.3 \pm 0.136$    | $83.4 \pm 0.065$    |
> | p-value   | -    | $1.24 \times 10^{-15}$  | $1.40 \times 10^{-11}$  | $1.48 \times 10^{-12}$  |
>
>
> **Responses to the Questions**
>
> **1. “Why are the experiments limited to chest X-ray data?...”**
>
> The research scope of this paper is on thorax disease classification based on chest X-rays. As stated in Section 1 of our paper, the image classification task on chest X-rays is more challenging than the image classification task on the general imaging domain since collecting a large dataset of high-quality annotations in medical domains is notably challenging or even impractical due to resource limitations or privacy issues. Generative Data Augmentation (GDA), which trains DNNs on augmented training data comprising both original training data in the standard benchmark datasets and synthetic training data generated by generative models such as Diffusion Models (DMs), shows the potential to mitigate the data scarcity issue. However, the synthetic data generated by GDA universally suffers from noise, and such synthetic data can severely hurt the performance of classifiers trained on the augmented training data. To address this issue, we propose a principled sample re-weighting method, Informative Data Selection (IDS), based on an established information-theoretic measure, the Information Bottleneck (IB), to improve the performance of DNNs trained for thorax disease classification with GDA. On the other hand, as a generic sample re-weighting method, IDS will be extended to the applications on other datasets for general machine learning and computer vision tasks.

---

### Meta-Review · Area_Chair_xEk6 · 2024-12-18

**Metareview:**

The paper’s introduction of the Informative Data Selection (IDS) method for re-weighting synthetic images in thorax disease classification was met with mixed reactions. While one of the reviewers found the method promising, the other reviewers identified key flaws in the paper's presentation, statistical rigour, computational efficiency, and practical relevance. The authors made substantial efforts to address these issues, conducting new experiments, improving clarity, and adding statistical significance tests. However, this was not enough to overcome the reviewers' criticisms. Some other issues:

-- The distinction between re-weighting and data augmentation was seen as inadequate by some reviewers. One reviewer suggested that IDS is, in essence, a data augmentation method.
-- Reviewers asked for a deeper discussion on how IDS could impact clinical practice and called for comparisons with methods focused on interpretability and uncertainty-based sampling.
-- Reviewers criticised the focus on thorax disease datasets and suggested extending IDS to other datasets. While the authors justified the focus, they did not demonstrate its generalisability.
-- Some reviewers felt that the performance improvements were too small to justify the complexity of IDS, despite the statistical significance

I would like to encourage the authors to improve their paper based on the feedback provided and resubmit to another venue.

**Additional Comments On Reviewer Discussion:**

Reviewer ntDN criticised the paper for its lack of statistical rigour in presenting results. They argued that the performance claims were not adequately supported due to the absence of statistical significance tests or estimates of variance. The reviewer also questioned the focus on thorax disease classification, suggesting the authors should evaluate the method on a broader range of datasets. Furthermore, they expressed scepticism regarding the effectiveness of IDS, asserting that randomness may have contributed to the observed performance gains. The authors conducted a series of t-tests on classification results, reporting p-values for comparisons with baseline methods to demonstrate that the improvements were statistically significant. They defended the choice to focus on thorax disease classification, highlighting the unique challenges posed by the medical domain and the potential applicability of IDS to other datasets in future work.


Reviewer Q4Pt raised concerns about the conceptual framing of the paper. While the authors framed IDS as a re-weighting method, the reviewer argued that it should be classified as a data augmentation technique, and therefore, it should be compared to other augmentation approaches. They also criticized the small performance gains (1-2% in classification metrics), which they felt were insufficient to justify the method's novelty. Additionally, they suggested exploring re-weighting for real data (not just synthetic) and questioned the relevance of localisation results for cardiomegaly, as bounding boxes around the heart would naturally overlap. The authors restructured the comparative analysis, adding more direct comparisons with active learning and data augmentation methods, as well as new experiments involving re-weighting real data. They also defended the relevance of localisation on cardiomegaly, arguing that it illustrates how IDS prioritises high-quality synthetic data. Moreover, they conducted statistical significance tests to demonstrate that the observed performance gains were not due to random chance.


Reviewer iQBZ was more supportive of the paper, noting that the IDS approach addresses a clear gap in synthetic data re-weighting for classification. They praised the paper's potential impact on clinical applications and commended the authors for providing access to the code, and enhancing reproducibility. However, the reviewer identified several areas for improvement, e.g. the clarity of Section 3.2, which contained dense formulas, the lack of an explanation for the metric mAUC, and concerns about the computational efficiency of the method, particularly with high-resolution images. The authors revised Section 3.2 and also clarified the definition of mAUC and added an efficiency analysis that detailed the training times of different model variants. The authors showed that IDS only slightly increased training time compared to baseline models while achieving better accuracy.


Reviewer Y6VF appreciated the conceptual relevance of IDS, especially its grounding in the Information Bottleneck (IB) principle. However, they criticized the paper for lacking methodological details, including gradient computation, centroid updates, and hyperparameter tuning. They also highlighted concerns about computational complexity, as diffusion models are known to be computationally expensive. The reviewer called for more comprehensive ablation studies to isolate the contributions of IDS's components and requested a discussion on the clinical relevance of the approach. The authors provided additional technical details on gradient computation and centroid updates, along with an explanation of their bi-level optimization framework. New ablation studies were added to isolate the contributions of the VIB and re-weighting network, which were tested on the COVIDx dataset. Furthermore, the authors reported on computational efficiency, showing that IDS introduces only a minor training overhead. However, the clinical relevance of IDS was not discussed as deeply as the reviewer had requested.

---

### Decision · Program_Chairs · 2025-01-22

Reject